# Genetic regulation of post-translational modification of two distinct proteins

Arianna Landini [1,9], Irena Trbojević-Akmačić [2,9], Pau Navarro [3], Yakov A. Tsepilov [4,5], Sodbo Z. Sharapov [4], Frano Vučković [2], Ozren Polašek[6,7], Caroline Hayward [3], Tea Petrović[2], Marija Vilaj[2], Yurii S. Aulchenko [4], Gordan Lauc[2,8,9], James F. Wilson [1,3,9 ✉] & Lucija Klarić [3,9 ✉]

Post-translational modifications diversify protein functions and dynamically coordinate their signalling networks, influencing most aspects of cell physiology. Nevertheless, their genetic regulation or influence on complex traits is not fully understood. Here, we compare the genetic regulation of the same PTM of two proteins – glycosylation of transferrin and immunoglobulin G (IgG). By performing genome-wide association analysis of transferrin glycosylation, we identify 10 significantly associated loci, 9 of which were not reported previously. Comparing these with IgG glycosylation-associated genes, we note protein-specific associations with genes encoding glycosylation enzymes (transferrin - *MGAT5*, *ST3GAL4*, *B3GAT1*; IgG - *MGAT3*, *ST6GAL1*), as well as shared associations (*FUT6*, *FUT8*). Colocalisation analyses of the latter suggest that different causal variants in the FUT genes regulate fucosylation of the two proteins. Glycosylation of these proteins is thus genetically regulated by both shared and protein-specific mechanisms.

[1] Centre for Global Health Research, Usher Institute, University of Edinburgh, Edinburgh, United Kingdom. [2] Genos Glycoscience Research Laboratory, Zagreb, Croatia. [3] MRC Human Genetics Unit, Institute for Genetics and Cancer, University of Edinburgh, Edinburgh, United Kingdom. [4] Laboratory of Glycogenomics, Institute of Cytology and Genetics, Novosibirsk, Russia. [5] Laboratory of Theoretical and Applied Functional Genomics, Novosibirsk State University, Novosibirsk, Russia. [6] Department of Public Health, School of Medicine, University of Split, Split, Croatia. [7] Algebra University College, Zagreb, Croatia. [8] Faculty of Pharmacy and Biochemistry, University of Zagreb, Zagreb, Croatia. [9] These authors contributed equally: Arianna Landini, Irena Trbojević-Akmačić, Gordan Lauc, James F. Wilson, Lucija Klarić. ✉ email: jim.wilson@ed.ac.uk; lucija.klaric@ed.ac.uk

Posttranslational modifications (PTMs) are essential mechanisms used by cells to diversify and extend their protein functions beyond what is dictated by protein-coding sequences in the genome. These chemical reactions range from the addition of small moieties, such as phosphate (phosphorylation), complex biomolecules, as in glycosylation, to proteolytic cleavage[1]. PTMs alter the structure and properties of proteins and are thus involved in the dynamic regulation of most cellular events. It is common for a PTM enzyme to target multiple substrates or interact with multiple sites. For example, only 18 histone deacetylases target more than 3600 acetylation sites on 1750 proteins[2]. Environmental or pathological conditions can lead to dysregulation of PTM activities, which has been related to aging[3] and several diseases, including cancer, diabetes, and neurodegeneration[4–6]. Despite their importance, little is known about the genetic regulation of posttranslational modifications.

N-glycosylation is one of the most common protein PTMs, where carbohydrate structures called glycans are covalently attached to an asparagine (Asn) residue of a polypeptide backbone. N-glycans are characterised by vast structural diversity and high complexity. While polypeptides are encoded by a single gene, N-glycan structures result from a sophisticated interplay of glycosyltransferases, glycosidases, transporters, transcription factors and other proteins[7]. Protein N-glycosylation is involved in a multitude of biological processes[8]. Accordingly, changes in N-glycosylation patterns have been associated with aging[9] and a wide range of diseases, including Parkinson's disease[10], lower back pain[11], rheumatoid arthritis[12], ulcerative colitis[13], Crohn's disease[13], type 2 diabetes[14] and cancer[15–17]. However, for most of these conditions, it still remains to be clarified whether the disease causes changes in N-glycosylation or vice-versa. In addition, N-glycans are considered as potential therapeutic targets[18] and prognostic biological markers[14,19–21].

As with other PTMs, genetic regulation of N-glycosylation is not yet fully understood. Previous genome-wide association studies (GWAS) have so far focused either on the N-glycome of total blood plasma proteins as a whole or on glycosylation of one specific protein—immunoglobulin G (IgG)[22–29]. IgG antibodies are one of the most abundant proteins in human serum, and their alternative N-glycosylation is suggested to trigger a different immune response and thus impacts the action of the immune system[30]. N-glycan structures are predominantly of the biantennary complex type and vary due to additions of core fucose, galactose, sialic acid and bisecting N-acetylglucosamine (GlcNAc), with disialylated digalactosylated biantennary glycan with core fucose and bisecting GlcNAc being the most complex N-glycan structure on IgG[31]. While a clear overlap in genetic control between total plasma proteins and IgG N-glycosylation was highlighted by previous studies[24], it was not possible, until recently, to identify protein-specific N-glycosylation pathways for glycoproteins other than IgG due to technical challenges of their isolation in large cohorts.

Here we investigate whether the same PTM of two proteins is regulated by the same genes and whether they are driven by the same causal genetic variants. We report genes associated with the regulation of transferrin N-glycosylation and compare these with the genetic regulation of glycosylation of a different protein (IgG). Transferrins are blood plasma glycoproteins regulating the level of iron in an organism. Iron plays a central role in many essential biochemical processes of human physiology: the cells' need for iron in the face of potential danger as an oxidant has given rise to a complex system that tightly regulates iron levels, tissue distribution, and bioavailability[32]. Human transferrin has two N-glycosylation sites—at the N432 and N630 residues, with biantennary disialylated digalactosylated glycan structure without fucose being the most abundant glycan attached[33,34]. We performed genome-wide association meta-analysis (GWAMA) of 35 transferrin N-glycan traits ($N = 1890$) and compared it with GWAMA of 24 IgG N-glycan traits ($N = 2020$) in European-descent cohorts, discovering both protein-specific and shared associations. For loci associated with the N-glycosylation PTM of both transferrin and IgG, we used colocalisation analysis to assess whether the underlying causal variants are protein-specific or rather shared between these proteins. We then suggested a molecular mechanism by which these independent causal variants could regulate the expression of glycosylation-related genes in different tissues.

## Results

**Loci associated with transferrin N-glycosylation.** To investigate the genetic control of transferrin N-glycosylation and assess whether the same genes and underlying causal variants are associated with N-glycosylation of both transferrin and IgG, we first performed GWAS of glycosylation for each protein (i.e. transferrin and IgG). A more extensive GWAS on the genetic regulation of IgG glycosylation has already been published[26], so we focus here on glycosylation of transferrin. We performed GWAS of 35 ultra-high-performance liquid chromatography (UHPLC)-measured transferrin N-glycan traits and Haplotype Reference Consortium (HRC) r1.1[35]—imputed genetic data in two cohorts of European descent, CROATIA-Korcula ($N = 948$) and VIKING ($N = 952$). Overall, we identified eight loci genome-wide significantly associated ($p$ value $\leq 1.43 \times 10^{-9}$) with transferrin N-glycans in the CROATIA-Korcula cohort (Supplementary Fig. 1 and Supplementary Data 1), six of which replicate in the VIKING cohort ($p$ value $\leq 0.00625$) (Supplementary Fig, 2 and Supplementary Data 2). Replicated loci contained genes encoding glycosyltransferases, enzymes directly involved in the biochemical pathway of N-glycosylation (MGAT5, ST3GAL4, B3GAT1, FUT8 and FUT6) and the transferrin (TF) gene. Cohort-specific heritability estimates for each transferrin glycan trait (Supplementary Data 3) ranged from 0% (VIKING TfGP2 and TfGP12) to 67% (CROATIA-Korcula TfGP23) and were high overall (>40% for the majority of the traits), similar to heritabilities previously reported for the total plasma glycome[36] as well as immunoglobulin G glycosylation[37]. To further increase the power of our analyses, we performed a fixed-effect inverse-variance meta-analysis of the discovery and replication cohort, discovering two additional loci (FOXI1 and HNF1A) (Table 1). To identify secondary association signals at each genomic region, we performed approximate conditional analysis on transferrin N-glycan traits using GCTA-COJO software[38]. Overall, we identified 15 independently contributing variants, located in ten genomic loci significantly associated ($p$ value $\leq 1.43 \times 10^{-9}$, Bonferroni adjusted for the number of glycan traits) with at least one of the 35 transferrin N-glycan traits (Table 1, Fig. 1 and complete list of all associations in Supplementary Data 4). Multiple SNPs independently contributed to transferrin N-glycan variation in four out of ten loci, all mapping to glycosyltransferase genes. The highest number of independently associated SNPs (3) was observed for the glucuronyltransferase locus, B3GAT1, while two SNPs contributed to transferrin N-glycan levels in the acetylglucosaminyltransferase locus (MGAT5), the fucosyltransferase locus (FUT8) and the sialyltransferase locus (ST3GAL4) (Supplementary Data 5).

To assess the potential impact of transferrin protein levels on the reported transferrin glycome associations, we utilised the transferrin *cis*-protein quantitative trait locus (pQTL), rs8177240[39]. This variant is associated with transferrin protein abundance and so can act as a proxy for protein levels and is not in linkage disequilibrium with glycan QTL (glyQTL) rs6785596, the sentinel glycosylation-

**Table 1 Loci genome-wide significantly associated with at least one of the 35 transferrin N-glycan traits in GWAMA.**

| Locus | Gene | SNP | EA | OA | EAF | No. of SNPs | Lead glycan | Phe. var. | No. of glycans | Beta | SE | P |
|---|---|---|---|---|---|---|---|---|---|---|---|---|
| 2:134839539-135024803 | MGAT5 | rs2442046 | C | G | 0.747 | 2 | TfGP23 | 0.071 | 4 | −0.44 | 0.037 | 1.38 × 10⁻³² |
| 11:126052988-126231874 | ST3GAL4 | rs405121 | T | C | 0.12 | 2 | TfGP17 | 0.131 | 9 | 0.782 | 0.046 | 9.67 × 10⁻⁶⁴ |
| 11:133906302-134613230 | B3GAT1 | rs74622686 | A | G | 0.905 | 3 | TfGP21 | 0.144 | 3 | 0.931 | 0.053 | 8.53 × 10⁻⁷⁰ |
| 14:65751627-66281192 | FUT8 | rs2411815 | A | T | 0.306 | 2 | TfGP20 | 0.092 | 3 | −0.469 | 0.035 | 2.69 × 10⁻⁴¹ |
| 19:5813766-5841356 | FUT6 | rs12019136 | A | G | 0.039 | 1 | TfGP32 | 0.079 | 5 | −1.016 | 0.083 | 2.00 × 10⁻³⁴ |
| 3:133433470-133499063 | TF | rs6785596 | A | T | 0.047 | 1 | TfGP3 | 0.065 | 3 | 0.787 | 0.075 | 1.57 × 10⁻²⁵ |
| 5:169535155-169535155 | FOXI1 | rs115399307 | T | C | 0.018 | 1 | TfGP23 | 0.031 | 1 | 0.941 | 0.152 | 5.18 × 10⁻¹⁰ |
| 8:15831868-16623073 | MSR1 | rs41341748 | A | G | 0.027 | 1 | TfGP35 | 0.031 | 1 | 0.778 | 0.109 | 1.16 × 10⁻¹² |
| 11:114381448-114384985 | NXPE1/NXPE4 | rs1671819 | A | G | 0.454 | 1 | TfGP14 | 0.02 | 1 | −0.2 | 0.032 | 3.32 × 10⁻¹⁰ |
| 12:121420263-121424861 | HNF1A | rs2393775 | A | G | 0.638 | 1 | TfGP28 | 0.019 | 1 | −0.203 | 0.033 | 8.97 × 10⁻¹⁰ |

Glycosyltransferase loci are reported at the top of the table, while other loci are listed at the bottom of the table. Each locus is represented by the SNP with the strongest association in the region, according to the p value rejecting the null hypothesis of no association with at least one of 35 transferrin glycan traits. An association was considered significant if the p value was lower than or equal to 1.43 × 10⁻⁹, the genome-wide significance threshold Bonferroni-corrected for the number of glycan traits. Locus—coded as 'chromosome: locus start-locus end' (GRCh37 human genome build); Gene—suggested candidate gene; SNP—variant with the strongest association in the locus; EA—SNP allele for which the effect estimate is reported; OA—other allele; EAF—frequency of the effect allele; No. of SNPs—number of SNPs in the locus independently contributing to trait variation according to GCTA-COJO; Lead glycan—glycan trait with the strongest association to the reported SNP; Phe. var.—proportion of variance in phenotype explained by the strongest associated SNP; No. of glycans—number of glycan traits significantly associated with variants at the given locus; Beta—effect estimate for the SNP and glycan with the strongest association in the locus; SE—standard error of the effect estimate; P—p value of the effect estimate (two-sided Wald test with one degree of freedom).

associated SNP in *TF* (LD $r^2 = 0.02$). Two glycans, TfGP3 and TfGP9, were significantly associated with transferrin *cis*-pQTL. However, both *cis*-pQTL and the glyQTL (rs6785596) contribute to the variation of TfGP3 levels, while only the *cis*-pQTL contributes to levels of TfGP9. Overall, this suggests that glycan associations with the *TF* gene are only completely accounted for by the transferrin protein levels in the case of TfGP9 (Supplementary Data 6). Further details about the potential effects of transferrin gene expression and protein levels can be found in Supplementary Results and Supplementary Data 19.

**Prioritising candidate genes associated with transferrin N-glycosylation.** For the ten loci associated with the transferrin N-glycome, we identified plausible candidate genes following multiple lines of evidence, such as evaluating the biological role of the candidate gene in the context of protein N-glycosylation, assessing colocalisation with eQTL, and investigating variant effects on the coding sequence or on putative transcription factor binding sites.

The majority of genes that were closest to variants associated with transferrin N-glycosylation had a clear biological link to protein N-glycosylation. In particular, for five out of ten loci, the closest genes (i.e. *MGAT5*, *ST3GAL4*, *B3GAT1*, *FUT8* and *FUT6*) encode glycosyltransferases, key enzymes in protein glycosylation, that have been previously associated with IgG and/or total plasma protein glycosylation (Supplementary Data 7). Another gene closest to variants associated with transferrin N-glycosylation and with a validated functional role in plasma protein glycosylation is *HNF1A*, a transcription factor previously associated with protein fucosylation (Supplementary Data 7). On the other hand, we also identified three loci that had not been associated with N-glycosylation. A locus on chromosome 3 contains the transferrin (*TF*) gene, which encodes the transferrin glycoprotein. A locus on chromosome 5 containing *FOXI1* encodes a member of the forkhead family of transcription factors (Forkhead box I1). Finally, a locus on chromosome 8 contains the *MSR1* gene, encoding the class A macrophage scavenger receptor, a trimeric integral membrane glycoprotein. Another gene of potential biological relevance at the chromosome 8 locus is the tumour suppressor candidate 3 (*TUSC3*), which encodes a protein localised to the endoplasmic reticulum and acts as a component of the oligosaccharyltransferase complex, responsible for N-linked protein glycosylation.

Using eQTL analysis in PhenoScanner, variants associated with transferrin N-glycosylation (and their proxies, LD $r^2 > 0.8$) were identified to be significantly associated with the expression of multiple genes in several human tissues involved in transferrin metabolism (Supplementary Data 8a). For example, variants associated with transferrin glycosylation were associated with *ST3GAL4* expression in liver and whole blood, with *B3GAT1* expression in the visceral adipose omentum, liver and whole blood, with *TF* expression in several adipose tissues and with *HNF1A*, *FUT8* and *MGAT5* expression in whole blood. The majority of these genes were also the closest to the strongest association in the locus. We next used Summary data-based Mendelian Randomisation (SMR) analysis followed by the Heterogeneity in Dependent Instruments (HEIDI) test[40] to assess whether expression of these genes colocalises with transferrin glycosylation (TfGP) traits. SMR-HEIDI provided evidence of colocalisation, suggesting that the same underlying causal SNPs are likely to regulate both transferrin glycosylation traits and gene expression, for *B3GAT1* in the liver and peripheral blood and *ST3GAL4* in the liver (Supplementary Data 8b).

We next explored whether any of the SNPs independently contributing to transferrin glycosylation (or their proxies) result

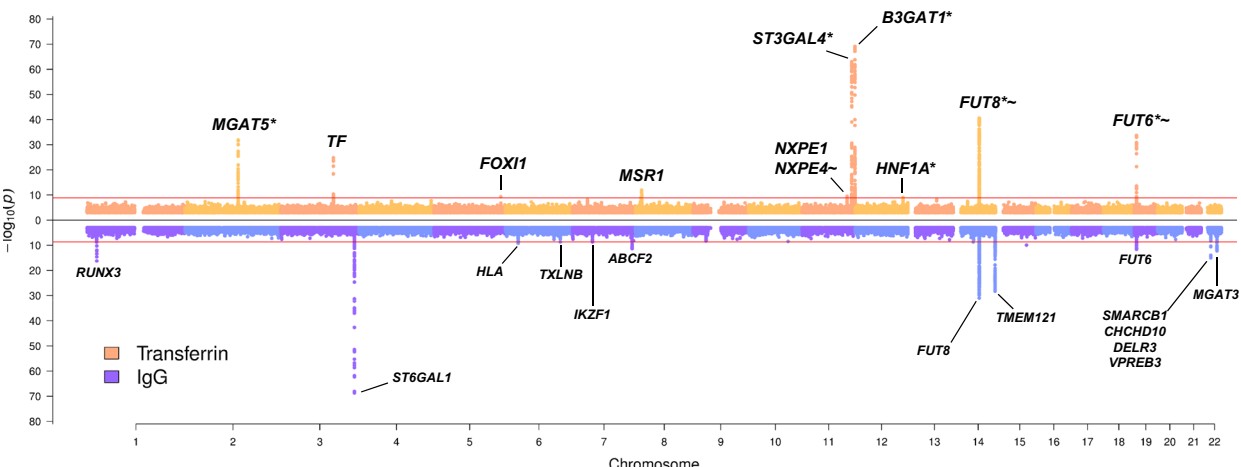

**Fig. 1 Transferrin and IgG N-glycome GWAMA summary Miami plot.** Miami plot pooling together meta-analysis results obtained across all 35 transferrin glycan traits at the top in orange, and across all 24 IgG glycan traits at the bottom in blue. The pooling was performed by selecting the lowest $p$ value (y-axis) from all 35 (TF) or 24 (IgG) glycan traits for every genomic position (x axis). For transferrin N-glycome associations, '*' marks loci previously reported in total blood plasma N-glycome GWAS[22-25], while '~' marks loci previously reported in IgG N-glycome GWAS[26-29]. The Bonferroni-corrected genome-wide significance threshold for the transferrin N-glycome meta-analysis (horizontal red line in the top part of the plot) corresponds to $1.43 \times 10^{-9}$, while Bonferroni-corrected genome-wide significance threshold for the IgG N-glycome meta-analysis (horizontal red line in the bottom part of the plot) corresponds to $2.08 \times 10^{-9}$. For simplicity, SNPs with $p$ value $> 1 \times 10^{-3}$ are not plotted. Gene or sets of genes annotated for transferrin N-glycome loci have been prioritised in this study; gene or sets of genes annotated for IgG N-glycome loci are those prioritised by Klarić et al.[26]. $P$ values are derived from the two-sided Wald test with one degree of freedom.

in a change of amino acid sequence using the Ensembl Variant Effect Predictor (VEP) v97[41]. While the majority of associated variants (>60%) were classified as intronic, several SNPs were identified as missense variants: rs115399307 (chr5:169535155-T/C) causes the substitution of the non-polar, aliphatic amino acid isoleucine (I) to the polar, hydrophilic amino acid threonine (T) in the FOXI1 transcription factor. Similarly, NXPE4 variant rs550897 (chr11:114442103-A/G, $r^2 = 0.94$ with rs1671819) causes an amino acid substitution from tyrosine (Y) to histidine (H). Genetic variant rs41341748 (chr8:16012594-A/G) disrupts a stop codon sequence in MSR1, causing an elongated transcript with the amino acid arginine (Arg) added to the protein chain (Supplementary Data 9). The FUT6 variant rs17855739 (chr19:5831840-T/C, $r^2 = 0.95$ with rs12019136) maps to the enzyme's catalytic domain and the allele T results in a change from negatively charged glutamic acid (E) to positively charged lysine (K), which leads to a full-length, but inactive, enzyme[42]. While the effect of reduced enzymatic activity on fucosylation of transferrin glycans needs to be experimentally validated, we observed that levels of TfGP32 are significantly lower in individuals carrying the T allele at rs17855739, compared to those with two C alleles (Supplementary Fig. 3). The structure of TfGP32 is currently not known, but its genetic association signal colocalises with two plasma glycan traits containing antennary fucose (A4F1G3S[3,3 + 6,3 + 6]3, A4F1G4S[3,3,3,6]4) and overall plasma antennary fucosylation (Supplementary Fig. 4 and Supplementary Results). Overall, this suggests that transferrin might contribute to these plasma glycan peaks and that TfGP32 might contain antennary fucose and could therefore be a proxy for FUT6 activity. However, these inferences need to be further experimentally validated.

Finally, we used the regulatory sequence analysis tools (RSAT)[43] to assess if variants associated with transferrin N-glycosylation overlap transcription factor binding sites and hence may be hypothesised to affect transcription factor binding. From the list of prioritised genes, we selected the two encoding transcription factors, FOXI1 and HNF1A, and checked whether associated variants in the remaining eight loci were likely to affect binding of

these transcription factors. Overall, the binding of both FOXI1 and HNF1A transcription factors might be affected by the sentinel variant (the SNP with the lowest $p$ value in the region for the given glycan trait) in the FUT8 gene. Similarly, the binding of HNF1A might be affected by the sentinel variants in the TF and ST3GAL4 loci (Supplementary Data 10).

**Shared genetic associations with complex traits and diseases.** To assess whether variants associated with transferrin glycosylation were also associated with complex traits and diseases we used PhenoScanner[44], followed by SMR-HEIDI to determine whether the shared associations are caused by the same underlying causal variant (colocalisation). We observed an overlap of transferrin N-glycan-associated SNPs and their proxies with variants associated with a complex trait- and disease-associated variants for five out of ten glycosylation loci (Supplementary Data 11a). For the remaining shared associations, we had no power to assess colocalisation (Supplementary Results for further details). Interestingly, variants at the TF locus have been previously associated with serum concentration of carbohydrate-deficient transferrins (CDT) (Supplementary Data 11a), less glycosylated transferrin isoforms traditionally used as a biomarker of excessive alcohol consumption[45], thus corroborating our finding for a related trait. We then assessed SMR-HEIDI findings (Supplementary Data 11b) using bi-directional Mendelian Randomisation (MR) to infer the causal direction between glycan traits and complex traits, and further validated the colocalisation results using a Bayesian approach. After Bonferroni correction ($p$ value $< 0.05/8 = 6.25 \times 10^{-3}$), there was no evidence of complex traits having an effect on glycan traits. However, we found positive associations of levels of TfGP14 and ulcerative colitis, and levels of TfGP28 and C-reactive protein levels, LDL and total cholesterol (Supplementary Data 12), although these results relied on few instrumental variables and were driven by associations in a single locus (Supplementary Fig. 5). Contrary to the SMR-HEIDI analysis, Bayesian colocalisation analysis suggested that the association of ulcerative colitis and TfGP14 levels at the NXPE1/NXPE4

locus are driven by independent, trait-specific causal variants. However, colocalisation confirmed that the associations between TfGP28 and C-reactive protein levels, LDL and total cholesterol are driven by a shared causal variant at the *HNF1A* locus (Supplementary Data 13 and Supplementary Fig. 5).

**Comparison of genetic regulation of glycosylation of transferrin and immunoglobulin G.** One of the main aims of this study is to understand if the N-glycosylation of two proteins is regulated by the same enzymes and if so, whether the same underlying genetic variant or a set of variants are driving the process. To address this question, in addition to the already described GWAMA of transferrin glycosylation, we performed a GWAMA of 24 UHPLC IgG N-glycan traits in the same individuals ($N = 2020$), following the same protocol. 13 loci were significantly associated with at least one of the 24 IgG N-glycan traits (Fig. 1 and Supplementary Data 14). The IgG N-glycome GWAS was annotated using genes or sets of genes prioritised by Klarić et al.[26] By comparing the two GWAS we discovered mainly protein-specific associations, but also two genomic regions that were associated with glycosylation of both proteins (Fig. 1). The protein-specific associations were with genes encoding known glycosylation enzymes (transferrin—*MGAT5*, *ST3GAL4*, *B3GAT1*; IgG—*ST6GAL1*, *MGAT3*), but also with transcription factors (transferrin—*HNF1A*, *FOXI1*; IgG— *IKZF1*, *RUNX3*), the protein itself (transferrin—*TF*; IgG—*TMEM121*, gene in the proximity of *IGH* genes encoding immunoglobulin heavy chains) as well as other genes (transferrin—*MSR1*; IgG—*TXLNB*, *ABCF2*, *SMARCB1* region, HLA-region). Interestingly, the regions containing *FUT8* and *FUT6*, genes encoding fucosyltransferases, enzymes adding core and antennary fucose, respectively, to the synthesised glycan, were associated with glycosylation of both proteins (Fig. 1). We then proceeded to assess whether the same underlying causal variants in these regions are controlling glycosylation for both proteins using colocalisation analysis.

Given that multiple glycan traits of the same protein can be associated with the same locus, we first asked whether all glycan traits of the same protein associated with a certain locus colocalise (Supplementary Fig. 6). Indeed, we found strong support for colocalisation (PP.H4 >80%, where PP.H4 represents the posterior probability for the same underlying causal variant contributing to trait variation), suggesting that for a given protein, all glycan traits associated with these loci are regulated by the same underlying causal variant (Supplementary Data 15 and Supplementary Figs. 7–9). One example of within-protein colocalisation can be seen in Fig. 2. We next tested whether, at the same genomic region, glycosylation of two different proteins is regulated by the same underlying causal variants. For this, we selected as the protein-representative glycan trait the one with the lowest *p* value in the given region (one pair for each locus— transferrin TfGP20 and IgG GP7 for the *FUT8* locus and transferrin TfGP32 and IgG GP20 for the *FUT6* locus) and proceeded to test for colocalisation between glycosylation of the two proteins. We found a strong support against colocalisation in both genomic regions (PP.H3 = 100% at *FUT8* locus, PP.H3 = 99.71% at *FUT6* locus, where PP.H3 represents the posterior probability for different underlying causal variants contributing to trait variation) (Figs. 3, 4 and Supplementary Data 16). Since colocalisation methods are sensitive to multiple independent variants in the region contributing to the trait variation, which was the case here, we validated our findings with the PwCoCo approach[46] (Methods) and again, obtained robust evidence against the colocalisation hypothesis for all tested traits in both loci (Supplementary Data 16 and Supplementary Results for further details).

Having established that different underlying causal variants regulate glycosylation at the *FUT6* and *FUT8* loci, we next explored the potential mechanisms behind these associations. The RSAT analysis suggests that the sentinel transferrin glycosylation SNP in the *FUT8* region might be affecting the binding of the HNF1A transcription factor (Supplementary Data 10). Similarly, it was previously shown that the sentinel IgG glycosylation SNP in the same *FUT8* region potentially affects the binding of the IKZF1 transcription factor[26]. In addition, we observed protein-specific associations with two transcription factors: transferrin glycosylation was associated with variants in the *HNF1A* locus and IgG glycosylation was associated with variants in the *IKZF1* locus (Fig. 1). We, therefore, checked the expression of these genes in tissues where the two proteins are predominantly expressed. It is known that plasma transferrin, encoded by *TF* gene, is mostly secreted by hepatocytes[47], while IgG, the heavy chain constant region of which is encoded by the *IGHG* gene, is predominantly synthesised by the antibody-secreting plasma cells, the fully differentiated form of B-lymphocytes[48]. Indeed, we see that *IGHG1* (encoding the most prevalent IgG1 subclass) is highly expressed in plasma cells and has low expression in hepatocytes, while the converse is true for *TF* (Fig. 5). Similarly, the transcription factor encoded by *HNF1A* is predominantly expressed in the hepatocytes, while *IKZF1* is mainly expressed in plasma cells (Fig. 5). Altogether these suggest that two distinct causal variants regulating glycosylation of transferrin and IgG in the *FUT8* locus might have tissue-specific effects, where the transferrin-associated variant affects the binding of HNF1A in the liver and the IgG-associated variant affects the binding of IKZF1 in plasma cells, with both influencing expressions of the *FUT8* gene and therefore affecting fucosylation of the two proteins.

## Discussion

Posttranslational modifications (PTMs) are essential mechanisms that dynamically regulate a large portion of cellular events by altering the structure and properties of proteins[1]. In common with other PTMs, genetic regulation of protein N-glycosylation has not been extensively investigated. Here, we performed a genome-wide association meta-analysis of glycosylation of two proteins—transferrin and IgG—and compared how their glycosylation is genetically regulated. In the GWAS of the transferrin N-glycome, ($N = 1890$), we identified ten significantly associated loci, two of which (near *FOXI1* and *MSR1*) were never previously associated with the glycome of any protein. The other eight have been previously associated with glycosylation of transferrin, total plasma proteins and/or IgG (Supplementary Data 7). The previous study on carbohydrate-deficient transferrin (CDT)[49], a composite measure that gives partial insight into the sialylation status of the protein, reported two genetic regions associated with the trait, near *PMG1* and *TF*, one of which we also found in this study (*TF*). Here, we were able to measure 35 different transferrin glycan traits, providing higher resolution of underlying structures and insight into the overall transferrin N-glycome. The total plasma glycome quantifies the glycome of all proteins in plasma, but without information on which glycan was bound to which protein. Given that IgG and transferrin are among the most abundant plasma glycoproteins[8], an overlap in genetic control of transferrin and IgG N-glycomes with that of total plasma proteins is to be expected. Sharapov et al.[24] previously indicated that some of the genomic loci associated with the plasma glycome overlap with loci associated with IgG N-glycosylation. The present work suggests that the *MGAT5*, *ST3GAL4* and *B3GAT1* loci, that were also observed in the total plasma protein GWAS, might be capturing a signal within plasma protein glycosylation that comes from transferrin N-glycosylation.

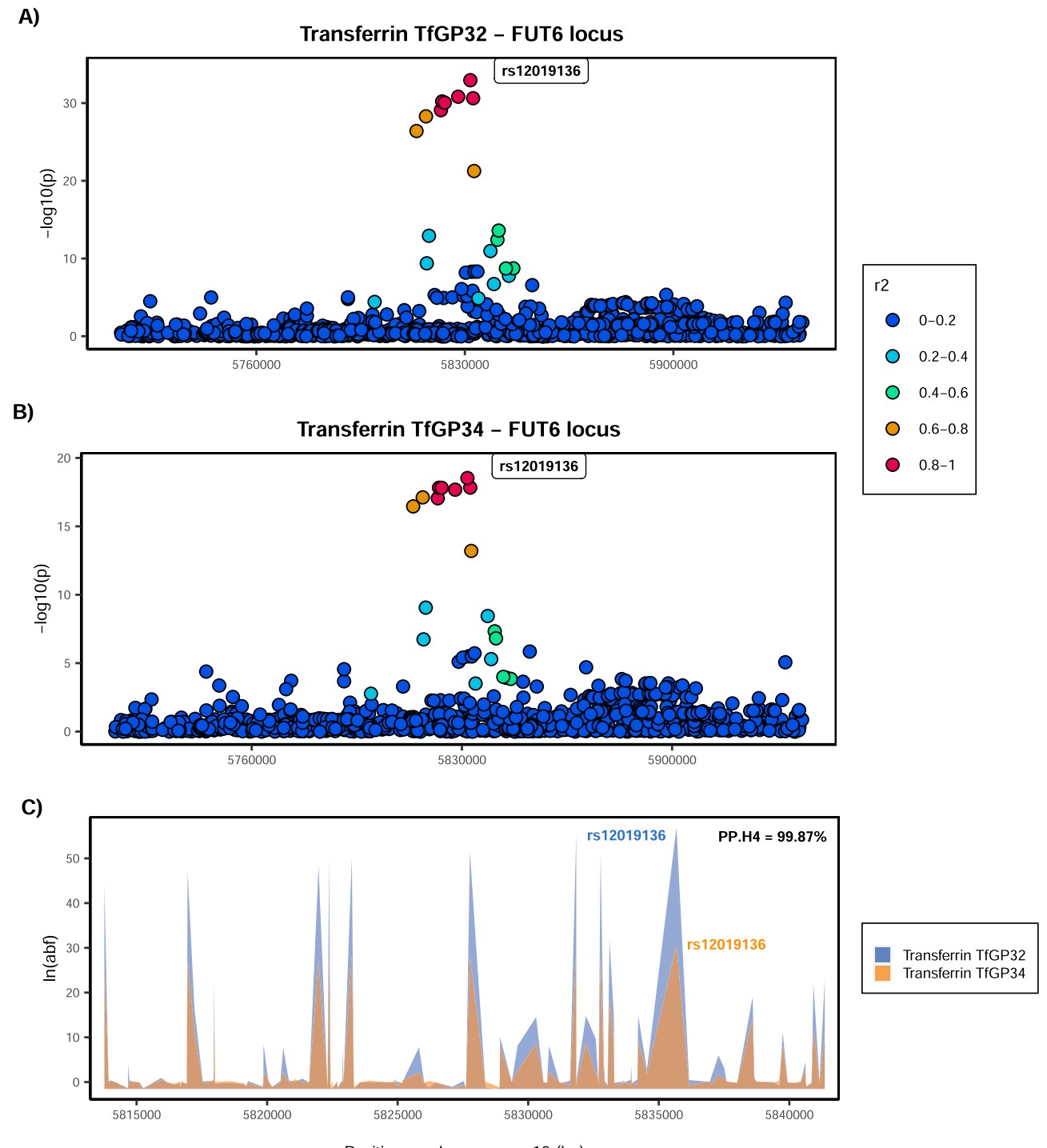

**Fig. 2 Colocalisation of two transferrin glycosylation traits in the *FUT6* locus (within-protein colocalisation).** Local association patterns for transferrin **A** TfGP32 and **B** TfGP34 glycans, and **C** their colocalisation pattern at the *FUT6* locus. TfGP32 and TfGP34 association patterns colocalise, with PP.H4 (posterior probability for hypothesis 4, of colocalisation) of 99.87%. The natural logarithm of Approximate Bayes Factor (ABF) of each SNP for transferrin TfGP32 and transferrin TfGP34 in the *FUT6* region shows that TfGP32 and TfGP34 associations are concordant (the patterns of ln(ABF) calculated for each SNP of both traits overlap), suggesting that the same underlying causal variant is associated with both traits. SNP most strongly associated in the region with the listed glycan trait is reported in bold and labelled.

We then compared the genetic architecture underlying glycosylation of transferrin and IgG proteins. Using the GWAS from this study we showed that there are both protein-specific and shared genetic loci. Looking specifically at glycosyltransferase enzymes, the main 'drivers' of this posttranslational modification, that catalyse the transfer of saccharide moieties from a donor to an acceptor molecule, *MGAT5*, *ST3GAL4* and *B3GAT1* were only associated with transferrin while *ST6GAL1* and *MGAT3* were only associated with glycosylation of IgG. On the other hand, two fucosyltransferase genes, *FUT8* and *FUT6*, were associated with both proteins. Since antennary fucose (produced by the FUT6 enzyme) is not typically found on IgG, we hypothesise that IgG

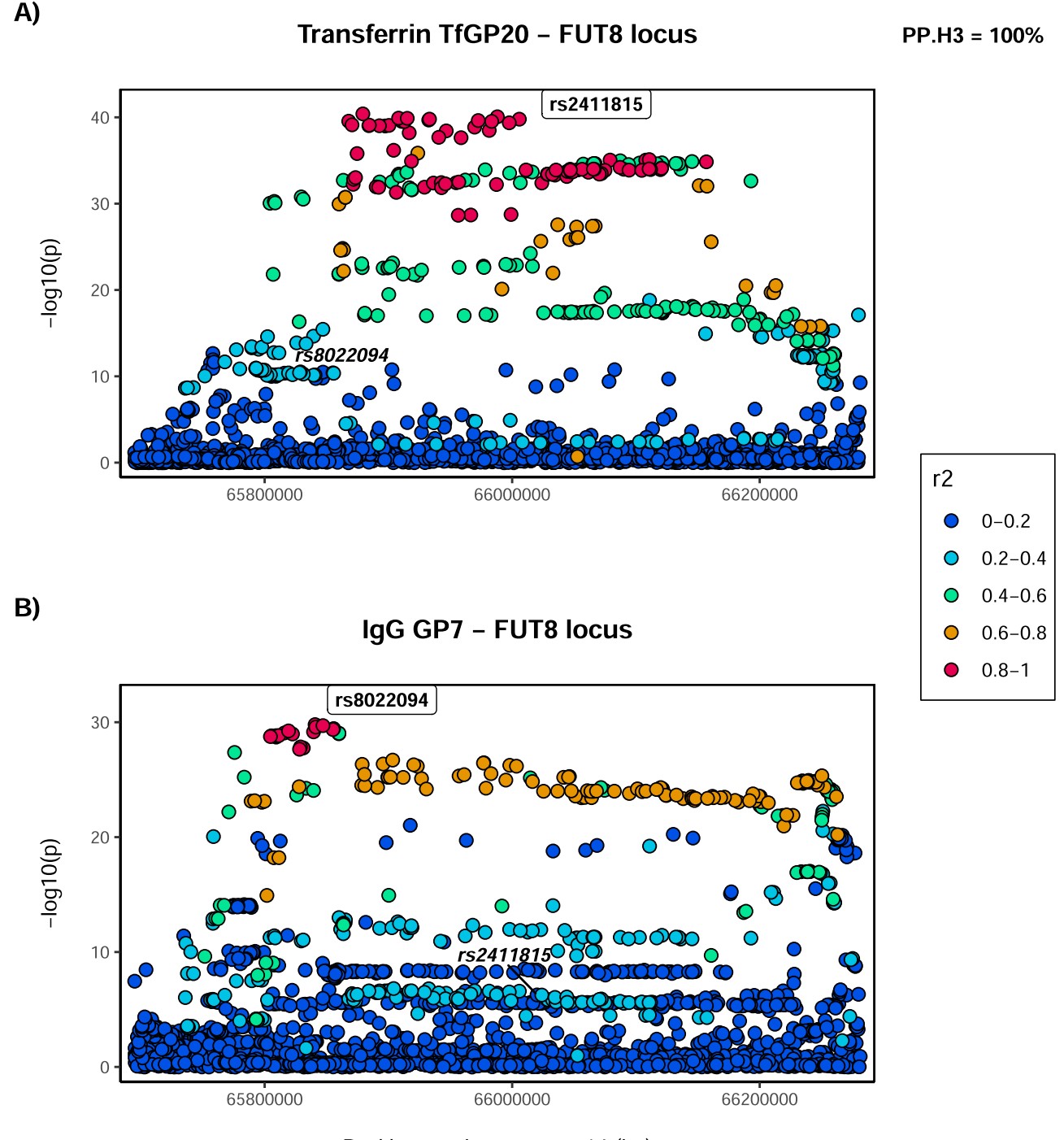

**Fig. 3 Colocalisation of transferrin and IgG glycosylation traits in the *FUT8* locus.** Local association patterns for **A** transferrin TfGP20 and **B** IgG GP7 glycans at the *FUT8* locus. TfGP20 and IgG GP7 association patterns do not colocalise, with PP.H3 (posterior probability for hypothesis 3, of different causal variants) of 100%. Colocalisation patterns are not reported since the width of the *FUT8* region makes the plot non-informative. SNP most strongly associated in the region with the listed glycan trait is reported in bold and labelled. For comparison, SNP most strongly associated with the other listed glycan trait is reported in italic, in the same panel.

glycosylation might be indirectly associated with *FUT6* through antennary fucosylation of other enzymes or proteins involved in glycosylation of IgG.

Even though the genes encoding FUT6 and FUT8 enzymes were associated with glycosylation of both proteins, using Approximate Bayes Factor colocalisation analysis, we showed that associations with transferrin and IgG N-glycosylation at these genomic regions are driven by independent underlying causal variants, where one variant regulates fucosylation of transferrin and the other of IgG. Our results suggest that while the same fucosyltransferase enzymes are involved in N-glycosylation of both transferrin and IgG proteins, the process is independently regulated by protein-specific causal variants.

There are at least two mechanisms that could explain how different variants in an enzyme-coding gene could have distinct

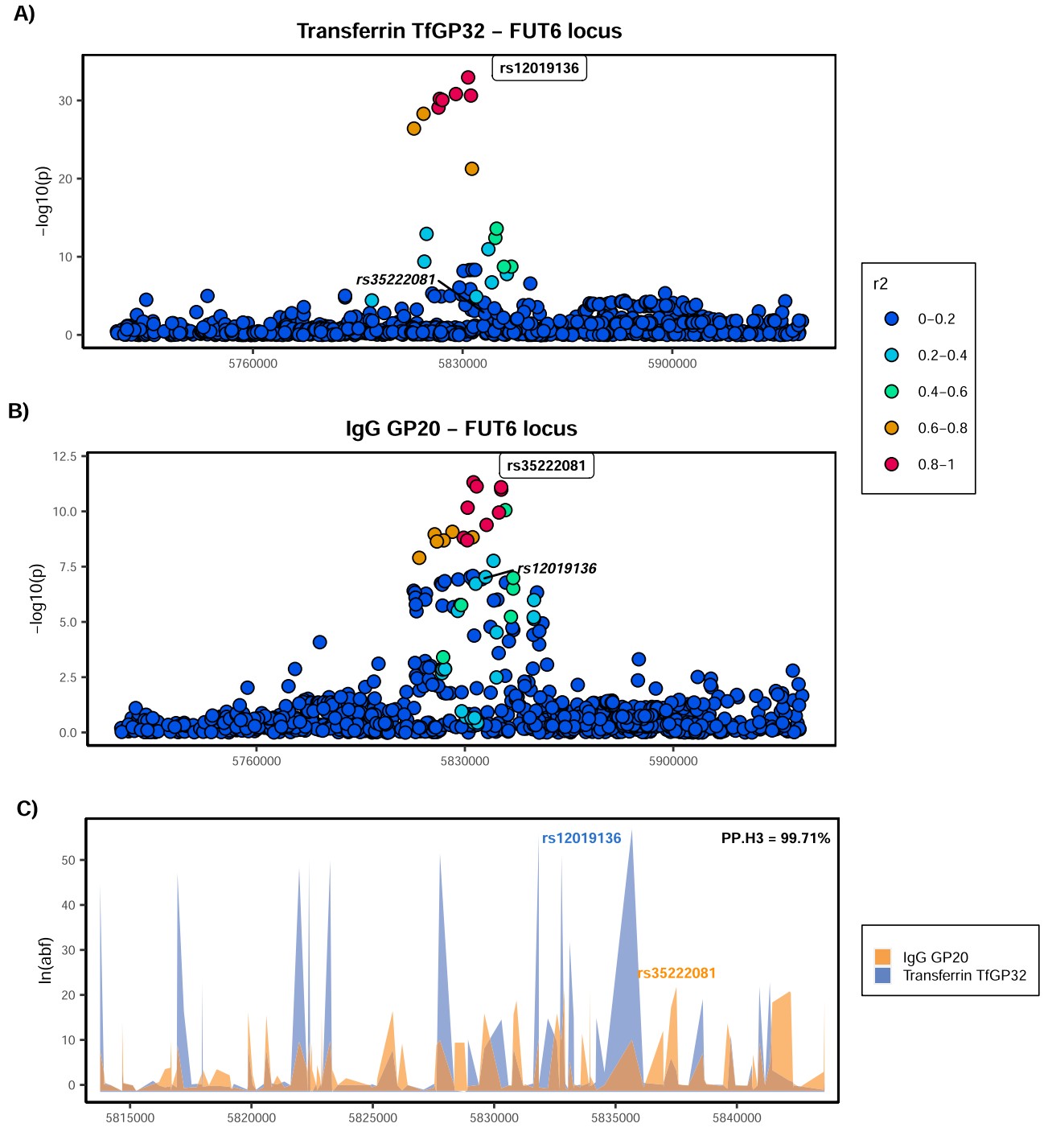

**Fig. 4 Colocalisation of transferrin and IgG glycosylation traits in the *FUT6* locus.** Local association patterns for **A** transferrin TfGP32 and **B** IgG GP20 glycans, and **C** their colocalisation pattern at the *FUT6* locus. TfGP32 and IgG GP20 association patterns do not colocalise, with PP.H3 (posterior probability for hypothesis 3, of different causal variants) of 99.7%. The natural logarithm of Approximate Bayes Factor (ABF) of each SNP for transferrin TfGP32 and IgG GP20 in the *FUT6* region shows that TfGP32 and GP20 associations are not concordant (the patterns of ln(ABF) calculated for each SNP of both traits do not overlap), suggesting that two different underlying causal variants in this region regulate glycosylation of these two proteins. SNP most strongly associated in the region with the listed glycan trait is reported in bold and labelled. For comparison, SNP most strongly associated with the other listed glycan trait is reported in italic, in the same panel.

effects on two different substrates. If the two variants were in the coding region of the gene and affected the amino acid sequence of the enzyme, they could affect the enzyme's specificity for binding each protein. However, the sentinel variant in the *FUT8* locus is not in strong linkage disequilibrium (LD) with coding variants

from the enzymes' active site, suggesting that this is likely not a common mechanism of regulation of fucosylation of the two proteins. In addition, overall, SNPs associated with transferrin glycosylation predominantly mapped to regulatory rather than coding regions of the genome (Supplementary Data 9). The other

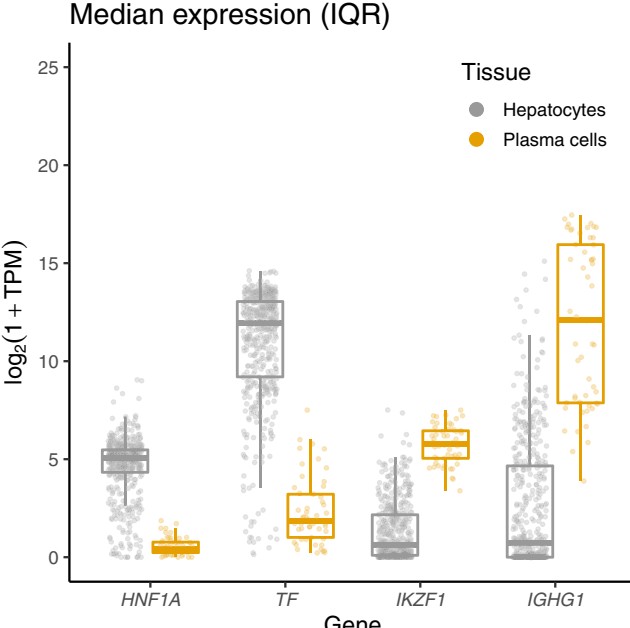

**Fig. 5 Expression of *TF, IGHG1, HNF1A* and *IKZF1* in main tissues of transferrin and IgG synthesis (liver and plasma cells).** Gene expression data, expressed in gene counts, was scaled to transcripts per million (TPM) and $\log_2(1 + \text{TPM})$ transformed. The data for hepatocyte ($N = 513$) and plasma ($N = 53$) cell samples were obtained from the ARCHS4 portal[72]. *TF* encodes transferrin protein, *IGHG1* encodes the constant region of immunoglobulin heavy chains, *HNF1A* and *IKZF1* encode two transcription factors involved in glycosylation of transferrin and IgG respectively. In the plot, the middle line represents the median, lower and upper limits of the box represent the first and third quartile, whiskers represent the 1.5 interquartile range. All individual data points are overlapped to the box plot.

hypothesis is that these two variants affect the expression of enzymes in different tissues. In common with all other antibodies, most of the IgG found in blood plasma is produced by bone marrow plasma cells, the fully differentiated form of B-cells[48]. The transferrin found in blood plasma is mostly produced by liver hepatocytes[47]. In addition, the glycomes of the two proteins were also associated with different transcription factor genes, namely, variants in the *IKZF1* region were associated with IgG glycosylation, and variants in the *HNF1A* region with transferrin glycosylation. IKZF1, a transcription factor predominantly expressed in immune cells and tissues, has been functionally validated as a regulator of IgG core fucosylation in lymphoblastoid cells: IKZF1 binds to regulatory regions of *FUT8* and, in turn, knockdown of *IKZF1* results in increased expression of *FUT8* and increased core fucosylation of IgG[26]. On the other hand, we showed that transferrin glycosylation-associated variants in the *FUT8* region might affect the binding of HNF1A, a transcription factor predominantly expressed in the liver. Lauc et al.[23] have shown that *HNF1A* knockdown results in downregulation of *FUT6* and upregulation of *FUT8* in the HepG2 hepatocyte cell line. While it might be expected that a change in levels of FUT6 and FUT8 enzymes would impact levels of antennary and core fucosylation (their enzymatic products), this link, especially in the context of transferrin glycosylation, has yet to be experimentally proven. Overall, our data could suggest that the two different causal variants may affect the binding of different transcription factors in different tissues and therefore regulate the glycosylation of the two plasma proteins in a tissue-specific manner. However, the effect of specific SNPs on the binding of the two transcription factors and their downstream effect on the expression of

fucosyltransferases in a tissue-specific manner still needs to be functionally validated.

In addition to HNF1A, variants in the *FUT8* locus associated with transferrin glycosylation might also be affecting the binding of the FOXI1 transcription factor. However, unlike HNF1A, a possible involvement of FOXI1 in the regulation of the transferrin fucosylation is to date unknown and would require functional validation. We also found that HNF1A binding could also be affected by variants associated with glycosylation in the *TF* and *ST3GAL4* genes. While these relationships were hitherto undocumented and need further supporting evidence, they may suggest that HNF1A might regulate multiple genes associated with transferrin N-glycosylation.

The most strongly N-glycosylation-associated variant for the *TF* gene, rs6785596, can be considered an example of a 'cis-glyQTL': a genomic locus that explains variation in glycosylation levels and is local to the gene encoding the protein being glycosylated. Similar was observed for IgG glycosylation, where associated variants mapped to the *IGH* locus[28], a genetic region encoding the heavy chain of immunoglobulin G. The transferrin glycosylation 'cis-glyQTL' is an eQTL for expression of transferrin in adipose tissue, but not in liver, where transferrin is predominantly expressed. The variant is also in middling LD ($r^2 = 0.57$) with a missense variant, rs179989, providing a potential alternative explanation for the association. Altogether, the exact mechanism of how these 'cis-glyQTL' could be affecting glycosylation levels remains unclear. Considering causal relations between the transferrin glycome and complex traits and diseases, we found associations between levels of TfGP28 and C-reactive protein levels, LDL and total cholesterol. These associations were, however, driven by a single locus encoding the transcription factor *HNF1A*, suggesting that the locus might be pleiotropic and has an impact on both transferrin glycan levels and complex traits.

In conclusion, by performing the GWAS of the plasma transferrin N-glycome and comparing it with that of the IgG N-glycome, we were able to describe similarities and differences in the genetic regulation of posttranslational modification of two different proteins. When focusing on glycosyltransferases, the main enzymes of this PTM, we showed that there are associations specific to each protein, but also those that are involved in the glycosylation of both proteins. For the latter, we showed that fucosylation of transferrin and IgG are regulated by independent, protein-specific variants in the *FUT8* and *FUT6* genes. In the *FUT8* region, these variants are likely to regulate fucosylation of transferrin and IgG in a tissue-specific manner, potentially acting through tissue-specific transcription factors. Additional studies, with larger sample sizes and focusing on other non-IgG proteins, will be necessary to further unravel the genetic architecture of N-glycosylation and to understand its relationship with human diseases and complex traits. While PTMs involved in intracellular signalling (e.g. phosphorylation) remain difficult to quantify in a high-throughput manner, here we investigated glycosylation of two plasma proteins, constraining the analysis to one type of PTM in the extracellular space. The impact of genetics on other, both intra- and extra-cellular posttranslational modifications will be an interesting area of future research.

## Methods

**Population cohorts.** The CROATIA-Korcula isolated population cohort includes samples of blood DNA, plasma and serum, anthropometric and physical measurements, information related to general health, medical history, lifestyle and diet for ~3000 residents of the Croatian island of Korčula[50]. Written informed consent was given and the study was approved by the Ethics Committee of the Medical School, University of Split (approval ID: 2181-198-03-04/10-11-0008). The Viking Health Study—Shetland (VIKING) is a family-based, cross-sectional study that seeks to identify genetic factors influencing cardiovascular and other disease risks

in the population isolate of the Shetland Isles in northern Scotland[51]. Genetic diversity in this population is decreased compared to mainland Scotland, consistent with the high levels of endogamy. A total of, 2105 participants were recruited between 2013 and 2015, most having at least three grandparents from Shetland. Fasting blood samples were collected and many health-related phenotypes and environmental exposures were measured in each individual. All participants gave written informed consent and the study was approved by the South East Scotland Research Ethics Committee, NHS Lothian (reference: 12/SS/0151). Details of cohort-specific demographics, genotyping, quality control and imputation performed before GWAS can be found in Supplementary Data 17.

**Phenotypic data.** Transferrin and total IgG N-glycome quantification for CROATIA-Korcula and VIKING samples was performed at Genos Glycobiology Laboratory. Isolation of the protein of interest and N-glycan quantification is described in more detail in Supplementary Materials and Methods and in Trbojević-Akmačić et al.[52] for transferrin and by Trbojević-Akmačić et al.[53] for IgG. Briefly, proteins were first isolated from blood plasma (IgG depleted blood plasma in the case of transferrin) using affinity chromatography binding respectively to anti-transferrin antibodies plates for transferrin and protein G plates for IgG. The protein isolation step was followed by enzymatic release and labelling of N-glycans with 2-AB (2-aminobenzamide) fluorescent dye. IgG N-glycans have been released from total IgG (all subclasses). N-glycans were then separated and quantified by hydrophilic interaction ultra-high-performance liquid chromatography (HILIC-UHPLC). As a result, transferrin and total IgG samples were separated into 35 (transferrin: TfGP1 − TfGP35) and 24 (IgG: GP1 − GP24) chromatographic peaks. It is worth noting that there is no correspondence structure-wise between transferrin TfGP and IgG GP traits labelled with the same number.

Prior to genetic analysis, raw N-glycan UHPLC data was normalised and batch corrected to reduce the experimental variation in measurements. Total area normalisation was performed by dividing the area of each chromatographic peak (35 for transferrin and 24 for IgG) by the total area of the corresponding chromatogram. Resulting measures are therefore relative abundances of each glycan structure in the overall glycosylation profile. Due to the multiplicative nature of measurement error and right-skewness of glycan data, normalised glycan measurements were $log_{10}$-transformed. Batch correction was then performed using the empirical Bayes approach implemented in the 'ComBat' function of the sva 3.34.0 R package[54], modelling the technical source of variation (96-well plate number) as a batch covariate. Batch corrected measurements were then exponentiated back to the original scale.

**Genome-wide association analysis.** Genome-wide association analyses (GWAS) were performed in the two cohorts of European descent, CROATIA-Korcula and VIKING. Associations with 35 transferrin N-glycan traits were performed in 948 samples from CROATIA-Korcula and 959 samples from VIKING. Associations with 24 IgG N-glycan traits were performed in 951 samples from CROATIA-Korcula and 1086 samples from VIKING. The sample size of the same cohort differs between transferrin and IgG due to the different number of samples successfully measured for each protein. Prior to GWAS, each glycan trait was rank transformed to normal distribution using the 'rntransform' function from the GenABEL 1.1-6 R package[55] and then adjusted for age and sex, as fixed effects, and relatedness (estimated as the kinship matrix calculated from genotyped data) as a random effect in a linear mixed model, calculated using the 'polygenic' function from the GenABEL R package[55]. Residuals of covariate and relatedness correction were tested for association with HRC (Haplotype Reference Consortium)[35] imputed SNP dosages using the RegScan v0.5 software[56], applying an additive genetic model of association.

**Meta-analysis.** Prior to meta-analysis, the following quality control was performed on cohort-level GWAS summary statistics. We removed all SNPs with a difference in allele frequency between the two cohorts higher than ±0.3 (~37,000 SNPs in total), as well as variants showing a minor allele count (MAC) lower or equal to 6 (~6 million SNPs in total). Cohort-level GWAS were then meta-analysed (N = 1890 for transferrin and N = 2020 for IgG N-glycans, for ~10.7 million SNPs) using METAL v2011-03-25 software[57], applying the fixed-effect inverse-variance method, followed by genomic control correction. The mean genomic control inflation factor ($\lambda_{GC}$) was 0.997 (range 0.982–1.011) for transferrin N-glycans and 0.995 (range 0.981–1.008) for IgG N-glycans meta-analysis, showing that the confounding effects of the family structure were correctly accounted for.

The standard genome-wide significance threshold was Bonferroni corrected for the number of N-glycan traits analysed: variants were considered statistically significant if their $p$ value was lower than $5 \times 10^{-8}/35 = 1.43 \times 10^{-9}$ for transferrin and $5 \times 10^{-8}/24 = 2.08 \times 10^{-9}$ for IgG N-glycan traits.

We used a positional approach to define genomic regions (loci) significantly associated with transferrin N-glycan traits, following the procedure adopted by Sharapov et al.[24]. For each glycan trait, we grouped all genetic variants located within a 500 kb window (±250 kb) from the sentinel SNP in the same locus. To obtain a unique list of loci that are independent of the specific glycan trait, we then merged this list of sentinel SNP-glycan trait pairs for all 35 glycan traits and applied

a similar procedure - all SNP-glycan trait pairs within a 1000 kb window (±500 kb from sentinel SNP) were grouped in the same locus, resulting in a unique list of sentinel SNP-top glycan trait pairs, summarising the genomic regions most strongly associated with N-glycans across all traits. A visual representation of the procedure can be seen in Supplementary Fig. 10. For all sentinel SNP-top glycan trait pairs, regional association plots were created with LocusZoom[58] and visually checked—in case of overlapping patterns of association, only the sentinel SNP-top glycan trait pair showed the lowest $p$ value was selected as a locus representative.

**Impact of transferrin protein levels on transferrin glycome associations.** To assess the potential impact of transferrin protein levels on transferrin glycome associations and to check whether the associations in the region of the *TF* gene are driven by protein levels, we tested the association of transferrin *cis*-pQTL rs8177240[39] with transferrin glycosylation using the likelihood ratio test implemented in the lmtest 0.9-38 R package[59] between the following models:

$$M0 : glycan \sim age + sex$$

$$M1 : glycan \sim age + sex + pQTL(rs8177240)$$

$$M2 : glycan \sim age + sex + glyQTL(rs6785596)$$

$$M3 : glycan \sim age + sex + pQTL(rs6785596) + glyQTL(rs8177240)$$

where pQTL is the SNP most strongly associated with transferrin levels and here used as proxy for the protein levels, and glyQTL is the SNP most strongly associated with transferrin glycan levels in the *TF* gene region.

The likelihood ratio tests were performed between:

- M0 and M1 to assess associations of glycans and pQTL (rs8177240)
- M1 and M3 to assess whether glyQTL contributes to glycan levels even when the pQTL is included in the model
- M2 and M3 to assess whether pQTL contributes to glycan levels even when the glyQTL is included in the model

To control for increased levels of relatedness between subjects in our studies, the models were fitted using linear mixed models as implemented in the lme4qtl 0.0.2 R package[60], with age, sex, pQTL and glyQTL as fixed effects and kinship matrix as a random effect. The kinship matrix was estimated from the genotyped data using the 'ibs' function from GenABEL[55] R package.

**Transferrin N-glycan traits post-meta-analysis follow-up.** The meta-analysis follow-up analyses were performed only for the transferrin N-glycans meta-analysis, since genetic regulation of IgG N-glycosylation has already been explored in a larger, IgG-specific study[26] and is beyond the scope of the present work.

**Conditional analysis and phenotypic variance explained.** To capture the overall contribution to phenotypic variation at each genomic region and identify secondary association signals at a locus, we performed approximate conditional analysis using the GCTA-COJO[38] 1.91.4beta stepwise model selection, 'cojo-slct', with the IgG and transferrin N-glycan meta-analysis summary statistics and genotypes of 10,000 unrelated individuals of white British ancestry from UK Biobank[61] as independent LD reference panel. Collinearity was restricted to 0.9 and the $p$ value threshold was set to $1.43 \times 10^{-9}$ for transferrin and to $2.08 \times 10^{-9}$ for IgG. Reported joint $p$ values were then adjusted by the genomic control method[62]. The list of samples for the independent LD reference panel was created with R 3.6.0, while the panel itself was generated using Plink 2.0[63]. After sample extraction from the UK Biobank full dataset, SNP deduplication was performed both by position (removing all SNPs not carrying a unique position on the chromosome) and marker name (--rm-dup exclude-all function). We acknowledge that UK Biobank might not be a perfect reference population for the CROATIA-Korcula cohort, however, there are no other reference panels with suitable ancestry and sample size (>4000)[38]. The proportion of variance (var) in phenotype (Y) explained by independently associated SNPs at each transferrin N-glycans associated locus was calculated with the following formula

$$var(Y) = \frac{2 * freq * (1 - freq) * \beta^2}{var(Y covariates adjusted residuals)} \quad (1)$$

where freq represents the frequency of the SNP's effect allele, $\beta$ is the effect estimate for the SNP and phenotype association at the locus, Y covariates adjusted residuals are the residuals resulting from the adjustment of the phenotype by age and sex, as fixed effects and relatedness (estimated as the kinship matrix calculated from genotyped data) as a random effect in a linear mixed model. The 'polygenic' function from the GenABEL R package was used also to estimate cohort-specific heritability for each transferrin glycan trait.

**Gene prioritisation.** For all genome-wide significant loci, we suggested plausible candidate genes combining different evidence, namely evaluating biological role in the context of protein N-glycosylation of genes nearest to sentinel variants

(positional mapping), assessing colocalisation with gene expression (expression quantitative trait loci, eQTL) or investigating associated variant's predicted effects on the protein sequence or on putative transcription factor binding sites. Positional gene mapping was performed using FUMA v1.3.5e SNP2GENE function[64]. Genes having a clear biological link to protein N-glycosylation (e.g. genes coding for enzymes involved in the biochemical pathway of protein glycosylation) and genes previously associated with IgG and/or total blood plasma proteins N-glycome were given a priority. The overlap of independent significant SNPs identified by COJO with eQTL was investigated using PhenoScanner v1.1 database[44], taking into account significant genetic association ($p$ value $< 5 \times 10^{-8}$) at the same or strongly (LD $r^2 > 0.8$) linked SNPs in populations of European ancestry. The Ensembl Variant Effect Predictor (VEP v97) tool[41] was used to determine putative functional effect and impact on a transcript or protein of independent significant SNPs and their strongly (LD $r^2 > 0.8$) linked SNPs in populations of European ancestry. Among genes prioritised so far, two were transcription factors (i.e. *HNF1A* and *FOXI1*), while the remaining were non-transcription factor protein-coding genes (i.e. *MGAT5*, *TF*, *MSR1*, *NXPE1/NXPE4*, *ST3GAL4*, *B3GAT1*, *FUT8* and *FUT6*). Using the Regulatory sequence analysis tools (RSAT) v2018-08-04 programme *matrix-scan*[43], we applied a pattern-matching procedure to search for sequences recognised as binding sites for *HNF1A* and *FOXI1* transcription factors in associated regions of the other eight prioritised genes. Position-specific scoring matrices (PSSMs), representing the frequency of each nucleotide at each position of the transcription factor motif, were downloaded for *HNF1A* and *FOXI1* from the JASPAR[65] database. For each of the eight genomic regions explored for possible transcription factor binding sites, we included the most strongly associated SNP and a 60 bp surrounding sequence (30 bp on either side of the sentinel SNP). The significance threshold was set to the $p$ value $\leq 0.003$, Bonferroni corrected for 16 tests performed (eight putative transcription factor binding sites tested for two transcription factors).

**Overlap and colocalization analysis with gene expression levels and complex traits**. The PhenoScanner v1.1 database[44] was used to investigate the overlap of significant transferrin glycosylation SNPs with gene expression levels and complex human traits. As previously described, we considered traits with genome-wide significant association ($p$ value $< 5 \times 10^{-8}$) at the same or strongly (LD $r^2 > 0.8$) linked SNPs in populations of European ancestry. We then used Summary data-based Mendelian Randomisation (SMR) analysis followed by the Heterogeneity in Dependent Instruments (HEIDI) test[40] to assess whether overlapping expression and complex traits, identified by PhenoScanner, were also colocalising with transferrin glycosylation (TfGP) traits. The SMR test indicates whether two traits are associated with the same locus, and the HEIDI test specifies whether both traits are affected by the same underlying functional SNP. Each of ten sentinel SNPs–TfGP pair (Table 1) was used for SMR/HEIDI analysis with gene expression levels and several complex traits. Summary statistics for gene expression levels in tissues/cell types were obtained from the Blood eQTL study[66] (http://cnsgenomics.com/software/smr/#eQTLsummarydata), the CEDAR project[67] (http://cedar-web.giga.ulg.ac.be/) and the GTEx project version 7[68] (https://gtexportal.org). Summary statistics for complex traits were obtained from various resources. In total, we used data for three tissues/cell types: CD19 + B-lymphocytes (CEDAR), GTEx liver (GTEx) and peripheral blood (the Blood eQTL study) and eight complex traits. A full list of GWAS collections, tissues and complex traits see in Supplementary Data 18. SMR/HEIDI analysis was performed according to the protocol described by Zhu et al.[40]. We used sets of SNPs having the following properties: (1) being located within ±250 kb from the sentinel SNPs identified in the present study; (2) being present in both the primary GWAS and eQTL data/GWAS for the complex trait; (3) having MAF ≥ 0.03 in both datasets; (4) having squared Z-test value ≥10 in the primary GWAS. Those SNPs that met criteria (1), (2), (3), (4), had the lowest $p$ value in the primary GWAS and were in high LD ($r^2 > 0.8$) with the sentinel SNPs were used as instrumental variables to elucidate the relationship between gene expression/disease and TfGP (we define them as 'top SNPs'). It should be noted that SMR/HEIDI analysis does not identify a causative direction affecting both traits. It can be either the top SNP or any other SNP in strong LD. After defining the set of eligible SNPs for each locus, we made the 'target' and 'rejected' SNP sets and added the top SNP to the 'target' set. Then we performed the following iterative procedure of SNP filtration: if the SNP from the eligible SNP set with the lowest PSMR had $r^2 > 0.9$ with any SNP in the 'target' SNP set, it was added to the 'rejected' set; otherwise, it was added to the 'target' set. The procedure was repeated until the eligible SNP set was exhausted, or the 'target' set had 20 SNPs. If we were unable to select three or more SNPs, the HEIDI test was not conducted. HEIDI statistics was calculated as

$$T_{HEIDI} = \sum_i^m z_{d(i)}^2, \qquad (2)$$

where m is the number of SNPs selected for analysis, $z_{d(i)} = d_i / SE_{(d_i)}$ and $d_i = \beta_{SMR_i} - \beta_{SMR(top\ SNP)}$.

The results of the SMR test were considered statistically significant if PSMR $< 1.7 \times 10^{-4}$ (0.05/302, where 302 is a total number of tests corresponding to analysed loci and gene expression/disease traits). Inference of whether a functional variant may be shared between the TfGP and gene expression/disease were made based on the HEIDI test: $P_{HEIDI} \geq 0.001$ (possibly shared) and $P_{HEIDI} < 0.001$ (sharing is unlikely).

We then proceeded to further explore SMR-HEIDI significant findings using bi-directional Mendelian Randomisation (MR), as implemented in the

TwoSampleMR 0.5.6 R package[69]. MR uses genetic variants as instrumental variables to investigate the effects of one trait (exposure) on another trait (outcome), assuming that the instrumental variables associate with the outcome only through exposure. GWAS summary statistics for complex traits were obtained from the IEU GWAS database[70] and their references are listed in Supplementary Data 18. For each glycan and complex trait, we selected as instruments for the exposure genetic variants associated with the trait at genome-wide significance ($p$ value $< 5 \times 10^{-8}$) and independent ($r^2 = 0.001$, using the European population from the 1000 Genomes Project reference panel). To distinguish causal relationships from confounding by LD, we followed up significant MR tests ($p$ value $\leq 0.05/8 = 6.25 \times 10^{-3}$, Bonferroni corrected for the number of tests) with approximate Bayes factor colocalisation analysis, developed by Giambartolomei et al.[71] and implemented in the 'coloc.abf' function from the coloc 4.0-6 R package, using default priors of $10^{-4}$ for the prior probability of SNP being associated with trait 1 or trait 2 (p1 and p2) and $10^{-5}$ for the prior probability of an SNP being associated with both traits (p12). To further assess the robustness of our findings, where available, we performed the 'coloc' analysis using a different complex-trait GWAS dataset compared to the SMR-HEIDI analysis (listed in Supplementary Data 18). Colocalisation analysis tests whether local genetic association signals for different traits are driven by the same shared causal variant or distinct variants. This Bayesian method provides posterior probabilities (PP) for five different hypotheses: the null hypothesis of no association with either of the traits (H0) and four alternative hypotheses of either association with only the first or the second of the traits (H1, H2), or association of both traits via distinct underlying causal variants (H3), or association of both traits through a shared causal variant (H4) i.e. trait colocalisation. A posterior probability >80% was considered as robust evidence supporting the tested hypothesis.

**Colocalisation analysis for transferrin and IgG N-glycan traits**. The *FUT8* and *FUT6* genomic regions were significantly associated with both transferrin and IgG N-glycans. To investigate a possible overlap in genetic control of glycosylation between the two proteins, we used the approximate Bayes factor colocalisation analysis implemented in coloc R package[71], followed by pairwise conditional and colocalization analysis (PwCoCo)[46] in case of multiple independent variants contributing to the trait variation. A posterior probability >80% was considered as robust evidence supporting the tested hypothesis.

Overview of the overall procedure can be seen in Supplementary Fig. 6. First, we assessed whether for one protein all glycans that are associated with the same genomic region ($p$ value $\leq 5 \times 10^{-8}$) are regulated by the same underlying variants. For each protein (i.e. transferrin and IgG) and each genomic region (i.e. *FUT8* and *FUT6*), we tested separately the group of glycans carrying only one independent association signal at the locus and the group of glycan traits showing multiple independent signals of association (Supplementary Fig. 6). Pairs of glycan traits obtaining a PP.H4 >80% (suggestive of colocalisation) were pooled in the same colocalisation group, following the principle that if trait A colocalises with trait B and trait B colocalises with trait C, thus also trait A and trait C colocalise. For each within-protein colocalisation group identified, the glycan trait with the lowest $p$ value was selected as a group representative and carried on to the next step, where traits with single and multiple independent associations for each protein were tested for colocalisation. Similar to previous steps, glycan traits were grouped together on the basis of their colocalisation analysis results and the lowest $p$ value representative was chosen for the next step, where finally representative transferrin and IgG glycans were tested for between-protein colocalisation.

For glycan traits with multiple independent association signals and lacking strong evidence for colocalisation, we applied the PwCoCo[46] approach. Briefly, the PwCoCo approach tests not only the traits' full, complete GWAS association statistics for colocalisation, but also summary statistics conditioned for the top primary association, testing whether any of the underlying causal variants between traits colocalise. For example, assuming that each trait is carrying two conditionally independent association signals in the tested region, colocalisation analysis will be conducted between both full and conditioned association statistics (conditioned for each independent variable), for a total of nine pairwise combinations. Secondary association signals at *FUT8* and *FUT6* loci for both transferrin and IgG N-glycans were assessed using GCTA-COJO approximate conditional analysis stepwise model selection[38] and an LD reference panel of 10,000 unrelated, white British ancestry individuals from UK Biobank[61]. We then performed the association analysis conditional on identified secondary association signals at *FUT8* and *FUT6* loci using GCTA-COJO[38] 'cojo-cond' and the same 10,000 UK Biobank samples LD reference panel, with $5 \times 10^{-8}$ $p$ value threshold and used those for pairwise colocalisation analyses.

**Expression of N-glycome associated genes in transferrin and IgG-relevant tissues**. Gene expression data for *TF, IGHG1, HNF1A* and *IKZF1*, expressed in gene counts, for hepatocytes (529 samples) and plasma cells (648 samples) was obtained from ARCHS4 portal[72]. Samples with a total number of gene count less than 5,000,000 were filtered out, leaving 513 hepatocyte and 53 plasma cell samples for the analysis. Gene counts were scaled to transcripts per million (TPM) and $\log_2(1 + TPM)$ transformed.

**Reporting Summary**. Further information on research design is available in the Nature Research Reporting Summary linked to this article.

## Data availability

The full summary statistics from the GWAS of 35 transferrin glycan traits and 24 IgG glycan traits generated in this study have been deposited in the DataShare repository. There is neither Research Ethics Committee approval, nor consent from individual participants, to permit the open release of the individual level research data underlying this study. The datasets analysed during the current study are therefore not publicly available. Instead, the research data and/or DNA samples are available from accessQTL@ed.ac.uk on reasonable request, following approval by the QTL Data Access Committee and in line with the consent given by participants. Each approved project is subject to a data or materials transfer agreement (D/MTA) or commercial contract. The UK Biobank genotypic data used in this study were approved under application 19655 and are available to qualified researchers via the UK Biobank data access process. The position-specific scoring matrices (PSSMs) for *HNF1A* and *FOXI1* genes used in this study are available in the JASPAR[65] database under the accession code MA0046.2 and MA0042.1, respectively. The summary statistics for gene expression levels in tissues/cell types used in this study are available in the Blood eQTL study, in the CEDAR project, in the GTEx project version 7 and in the eQTLGen consortium. The summary statistics for complex traits are available in various publicly available resources, as detailed in Supplementary Data 18.

## Code availability

The following software packages were used in this study: Ensembl variant effect predictor (VEP): phenoscanner: GCTA-COJO: coloc: TwoSampleMR: The remaining code used in this paper may be requested from the authors.

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

## Acknowledgements

We thank Dr Nicola Pirastu for sharing his knowledge and experience with Mendelian Randomisation and Jelena Šimunović for her technical assistance in the laboratory work. The CROATIA-Korcula study was funded by grants from the MRC (United Kingdom), European Commission Framework 6 project EUROSPAN (contract number LSHG-CT-2006-018947), Croatian Science Foundation (grant 8875) and the Republic of Croatia Ministry of Science, Education and Sports (216-1080315-0302). Genotyping was performed in the Genetics Core of the Clinical Research Facility, University of Edinburgh. We would like to acknowledge all the staff of several institutions in Croatia that supported the CROATIA-Korcula fieldwork, including, but not limited to, the University of Split and Zagreb Medical Schools, Institute for Anthropological Research in Zagreb, and the Croatian Institute for Public Health in Split. The Viking Health Study— Shetland (VIKING) was supported by the MRC Human Genetics Unit quinquennial programme grant 'QTL in Health and Disease'. DNA extractions and genotyping were performed at the Edinburgh Clinical Research Facility, University of Edinburgh. We would like to acknowledge the invaluable contributions of the research nurses in Shetland, the administrative team in Edinburgh and the people of Shetland. Finally, we thank the UK Biobank Resource, approved under application 19655. We acknowledge support from the European Union's Horizon 2020 research and innovation programme IMforFUTURE (A.L.: H2020-MSCA-ITN/721815); the RCUK Innovation Fellowship from the National Productivity Investment Fund (L.K.: MR/R026408/1); the Russian Science Foundation (RSF) (Y.S.A. and S.Z.S.: 19-15-00115); and the MRC Human Genetics Unit programme grant, 'QTL in Health and Disease' (J.F.W. and C.H.: MC_UU_00007/10).

## Author contributions

A.L.: Data analysis and interpretation, visualisation, writing—original draft preparation, writing—review and editing. I.T.-A.: Quantification of transferrin and IgG N-glycans, data interpretation, writing—original draft preparation, writing—review and editing. P.N.: Supervision, data interpretation, writing—review and editing. Y.A.T.: Data analysis and interpretation, writing—original draft preparation. S.Z.S.: Visualisation, writing—original draft preparation. F.V.: Glycan data quality control. O.P.: Genomic and demographic data provider for CROATIA-Korcula cohort. C.H.: Genomic and demographic data provider for CROATIA-Korcula cohort. T.P.: Quantification of transferrin and IgG N-glycans. M.V.: Quantification of transferrin and IgG N-glycans. Y.S.A.: Writing—review and editing. G.L.: Conceptualisation, glycan data provider for CROATIA-Korcula and VIKING cohorts, writing—review and editing. J.F.W.: Conceptualisation, genomic and demographic data provider for VIKING cohort, supervision, data interpretation, writing—original draft preparation, writing—review and editing. L.K.: Conceptualisation, supervision, data interpretation, writing—original draft preparation, writing—review and editing.

## Competing interests

G.L. is the founder and owner of Genos Ltd, a private research organisation that specialises in the high-throughput glycomic analysis and has several patents in this field. I.T.-A., F.V., T.P., and M.V. are employees of Genos Ltd. Y.S.A. is a founder and a co-owner of PolyOmica and PolyKnomics, private organisations providing services, research and development in the field of computational and statistical genomics. The remaining authors declare no competing interests.
