## [Peer Review File · Nature Communications]

Same role but different actors: genetic regulation of post-translational modification of two distinct proteinsReviewers' Comments:

Reviewer #1:

Remarks to the Author:

This manuscript by Landini et al describes the results of a GWA analysis of transferrin and IgG glycosylation with the goal to determine the genetic control of transferrin glycosylation as well as to determine whether the same genes regulate glycosylation on different proteins. The authors report that some glycosylation-associated genes influence the glycosylation of transferrin, but not IgG, while others are shared. The authors conclude that there are similarities and differences in the genetic control of post-translational modifications of different proteins and that these characteristics are relevant to understand their relationship with human diseases and complex traits.

Comments:

The authors made use of previously published data on IgG glycosylation as well as newly generated data obtained after analyses of the glycosylation profile of transferrin. In these studies, they performed a total glycan release from purified transferrin. It is important that transferrin is isolated to high purity as potential other proteins could also contribute to the glycan profile analysed. Therefore, a SDS-page silverstaining should be shown to access the purity of the transferrin fraction analysed.

The authors indicate that not all SNPs were considered in the meta-analyses of the two cohorts. It should be specified how many SNPs were removed prior to meta-analyses as this could potentially introduce bias.

The authors indicate that several SNPs associating with transferrin glycosylation are classified as missense variations causing amino acid substitutions. For example, it is indicated that the FUT6 variant encodes an amino acid change that leads to an inactive enzyme. If so, it is to be expected that alpha-1,3 fucose expression on transferrin/IgG is decreased. The conclusions presented by the authors would be substantiated considerably in case this is analysed specifically in donors expressing this variant as opposed to donors expressing two functional alleles.

If I am not mistaken, alpha1,3 fucose is hardly present in IgG1, and -as such- it is surprising to note that FUT6 associates with IgG N-glycosylation. Could the authors comment on this notion and indicate where and how often this fucose is present on IgG1.

As indicated above, the data conclusions would be substantiated considerably in case the authors would quantify this glycoform on IgG derived from relevant donors.

From the description it is not clear which IgG subclass is analysed. This should be specified.

The subsequent analyses assessing whether transferrin glycosylation variants associate with other complex traits add an additional complexity to the study and provide little mechanistic insight into the biology of the associations. These studies, presented in supplementary table 7, are also prone to false positive findings and should be replicated in an independent manner.

The authors show that IGH1 is, as expected, not expressed by hepatocytes, but expressed by plasma cells. The authors also show that these cells, but not hepatocytes, express IKZF1. Something similar is found for HNF1A and TF in hepatocytes. It is, however, unclear to me why this would suggest that variants of HNF1A/IKZF1 would be involved in regulation of FUT8 expression in liver/plasma cells without additional functional studies. Although, the authors published on IKZF1 and IgG glycosylation, such studies should be performed for hepatocytes as well to drive this case more convincingly.

Reviewer #2:

Remarks to the Author:

Review of “Same role but different actors: genetic regulation of post-translational modification of two distinct proteins” by Landini et al (Nature Comms).

Reviewer expertise: human genomics, QTL studies.

This study identifies genetic variants that are associated with glycosylation of two proteins: transferrin and IgG.

The work is important and novel; many large-scale studies have described the impact of genetic variation on protein abundance (pQTLs) in plasma (eg Surhe Nat Comms 2017, Folkersen Plos Genetics 2017, Sun et al Nature 2018, Folkersen Nat Metab 2020) but few have examined the effect of genetics on post-translational modifications (PTMs). PTMs are an important part of the proteomic landscape and improved understanding of factors influencing them is an important next step in proteogenomics.

The authors report two findings that I found particularly interesting and that have broader relevance beyond the study of the specific PTM and proteins examined here. First, they demonstrate examples of genetic variation in/near the genes encoding the target proteins (transferrin and IgG) that affect post-translational modification of said protein. Second, they show both distinct and shared sets of genetic variants acting on the same PTM (glycosylation) for different proteins (i.e. the genetic influences are not all non-specific/ a generic effect). This is nicely visualised with the Miami plot. They further expand on this observation by showing that even when the same locus is associated with glycosylation of transferrin and IgG, the underlying causal variants can differ (examples of *FUT6* and *FUT8* loci).

The manuscript was well-written and clear with high quality visualisation.

The analytical techniques were appropriate.

Comments and suggestions for revision (key comments indicated with **)

-Lines 51-54: various associations between N-glycosylation and diseases listed. The authors should spell out that these correlations are not necessarily causal (eg it seems unlikely that N-glycosylation is the cause of low back pain).

-Use of ‘SNP’ throughout the manuscript – do the authors really only mean SNP ie are none of the instances referring to indels at all (which will also be included with the array genotyping + imputation strategy they described)? If they actually mean SNPs + indels then they should change to ‘SNPs’ to “genetic variants”.

-Line 147-8: “transferrin glycosylation variants” – I dislike the terminology as it’s potentially misleading (the reader may think the variants are all in transferrin gene region). Please say “variants associated with transferrin glycosylation” instead.

** -Lines 151-157: the authors use the Heidi-SMR analysis to conclude the same variant underlies the glycosylation trait and the eQTL. It would be helpful to understand the causal relationship (or lack thereof) between the mRNA trait and the glycosylation trait, and the direction of any causal effect. ie does SNP -> trait A ->

trait B, or does SNP → trait B → trait A, or does SNP separately affect trait A and trait B, but not via each other? The paper would be strengthened by performing bidirectional MR / mediation analysis to tease this out.

A similar comment applies to line 193: the authors have a similar interesting link between a variant associated with ulcerative colitis (UC) and glycosylation. Extending this to formally test whether transferrin glycosylation plays a causal role in UC via MR would greatly strengthen the potential translational value of the paper.

** - How many of the individuals in the cohort had diabetes? Have the authors considered whether tendency to diabetes or “pre diabetes”/ relative insulin resistance (or even just different glucose levels with the continuum of normal) might have impacted their findings?

This could be addressed in a couple of ways: i) do the loci associated with glycosylation overlap with diabetes susceptibility loci (T2 probably most relevant given its prevalence) or loci associated with HbA1c levels? ii) If the authors have measures of glycaemia available (eg HbA1c contemporaneous with plasma sample), they could add this as a covariate to see whether this attenuates any of the loci? If so, they could do a formal mediation analysis to test the hypothesis that the effect is mediated via blood sugar levels at the relevant locus.

-Lines 159-171. The authors identify some instances where a protein-coding variant is in high LD with the variant associated with glycosylation. What was the situation for the ‘cis’ acting variant at *TF* (encoding transferrin itself)? If there is no protein-coding variant in high LD with the lead variant, then can the authors comment on how it might be impacting glycosylation (as transferrin protein structure would presumably not be affected?). Is this variant a cis eQTL or pQTL for *TF*?

Some additional comments arise from this:

** - Is glycosylation analysed as a quantitative measure or a ratio of glycosylated to unglycosylated? If the former, then simply increasing total protein abundance might lead to a higher amount of glycosylated protein without changing the ratio. The authors should clarify whether their approach has taken this into consideration as this will have an important bearing on the interpretation of results (analogous to looking at absolute vs proportional cell counts in a full blood count).

-Line 176 – ‘TF’ abbreviation for transcription factor. Please avoid this since it creates confusion with the *TF* gene discussed elsewhere!

-Figure 2A and 2B, and Figure 4A and B would be better presented vertically so the reader can more easily see how the signals compare.

-The authors rightly discuss the importance of PTMs in their many various contexts. The Discussion should highlight that this study is limited to plasma proteins i.e. extracellular space. Some of the PTMs discussed in the manuscript (e.g. phosphorylation) play an important role in intracellular signalling with dynamic on/off effects. The impact of genetics on these types of PTMs will be very challenging to assess, but will be an interesting area for future research.

-Lines 388-397: I found this a bit speculative even for Discussion and felt it would be better removed. Ultimately as the authors point out “However, this SNP has so far not been associated with diabetes or diabetes-related traits, suggesting that this relationship needs to be explored further”. Without this confirmatory link, the point is not particularly convincing.

-Lines 402-403 “glycosylation SNPs in NXPE1/NXPE4 locus were pleiotropic with ulcerative colitis” – I found the phrasing odd. I realise the terminology comes from SMR HEIDI but it is misleading language. A variant associated with a single disease and a molecular trait can hardly be considered pleiotropic in the usual sense. Disease-associated variants must have molecular underpinnings and so association with some molecular trait is inevitable (even if yet to be discovered). I would reserve the term ‘pleiotropic’ for when the variant is associated with multiple traits within the same trait type (eg with multiple proteins, or with multiple diseases). The authors should rephrase this more simply to say what they mean ie “variants in NXPE1/NXPE4 locus associated with glycosylation were also associated with ulcerative colitis”. There were a couple of other instances where pleiotropy was referred to that would similarly benefit from clearer language. The other reason for doing this is to avoid confusing readers more familiar with MR, where horizontal pleiotropy violates MR assumptions and thus calls into question the conclusions of an MR analysis.

-Line 502 Locus Definition. I found the explanation confusing. Clarifying or providing pseudocode or real code might help understand what was done in terms of grouping loci across different glycosylation traits.

** -To maximise utility of the results to the community, the authors should ensure full summary stats are uploaded to an appropriate repository (eg GWAS catalog).

Caveat

The transcription factor analysis was beyond the scope of my expertise and I do not feel qualified to judge this.

Reviewer #3:

Remarks to the Author:

Landini and coworkers investigate post-translational modifications of transferrin and immunoglobulin. In focus is the glycosylation of transferrin, which have not before been investigated in a GWAS. The study approach is overall well-done and methodology is according to standards for metabolic and intermediate trait GWAS. The idea of comparing glycosylation in the two proteins, using co-localization is good, and have the potential to create knowledge of how the process of glycosylation generally works.

It is at times difficult to follow the different glycan traits, because the data is virtually always collapsed into strongest association. This perhaps makes sense in the main text and tables, but particularly in supplementary tables, I recommend providing a complete overview of all glycan traits with all independent variants (i.e. all GCTA-COJO independent associations to all measured glycan traits), which could help with subsequent two points as well. Alternatively please provide some explanation or quantification of differences in glycan traits.

Please provide more details regarding the findings in each of the two individual cohorts, e.g. calculate a heterogeneity metrics or list effect size and P-value per cohort, and summarize this in main text. Supplementary figure 5 and 6 are very useful towards this, and supplementary table 13 also serves some of this purpose, but it is for example not possible to know all 10 findings of e.g. table 1. Also, it took me a while to figure out the high level of replication between the two cohorts, which is why I think it could be highlighted more.

Please provide calculations of proportion of variance explained for each association and discuss the overall proportion of variance explained by genetics for each trait. This is currently only provided for top-SNP per locus (table 1), not all independent SNPs. It will be of interest to readers to know to what degree these traits are genetically regulated overall. Also consider adding in LDSC calculations, or at least discussing possibility of low-effect background.

The study is based on isolated populations, which may be in some contexts be considered a strength. However, the GCTA-COJO needs a standardized LD value source, which is indicated as being from the UK biobank. Will this affect the results, in case the LD patterns are different? Was this tested?

At lines 105-109, there are 6 genes with independent contributions listed: ST4GAL4, MGAT5, B3GAT, FUT8, FUT6 and TF. The numbers listed are 7, 4, 4, 2, 2 and 2. In addition to the remaining 4 genes, adding up to 10 loci with significant variants. But that sums up to only 25 - not 26 independently contributing variants, stated in line 100. Please double-check that reported counts are correct.

The study by Kutalik et al 2011, PMID 21665994, "Genome-wide association study identifies two loci strongly affecting transferrin glycosylation" is related to this manuscript. The study is already briefly mentioned in supplementary table 7a, but the manuscript could benefit from more discussion and comparison with such a related study.

I think it would be more useful to readers with the very big supplementary tables in a more data-readable format than pdf (e.g. xlsx or txt).

In Supplementary Table 7a the first two columns are completely identical, except the header name; snp and ref_rsids. What's the motivation to provide that twice? If none, consider removing.

Reviewer #1 (Remarks to the Author):

The authors made use of previously published data on IgG glycosylation as well as newly generated data obtained after analyses of the glycosylation profile of transferrin. In these studies, they performed a total glycan release from purified transferrin. It is important that transferrin is isolated to high purity as potential other proteins could also contribute to the glycan profile analysed. Therefore, a SDS-page silverstaining should be shown to access the purity of the transferrin fraction analysed.

We thank the Reviewer for raising this important question. During purification procedure development, we saw that low amounts of IgG co-purify with transferrin in case when transferrin isolation is done directly from the plasma sample. However, when IgG is depleted and flowthrough used for subsequent transferrin isolation, there is no visible contamination from IgG in transferrin eluates by SDS-PAGE. Moreover, transferrin purification is further improved by using 1xPBS with increased concentration of sodium chloride as a washing buffer during purification. More details can be seen in Trbojević-Akmačić, I. *et al.*¹ To ensure high purity of isolated transferrin in this comparative study, we have first performed IgG isolation and then used the flowthrough to immediately isolate transferrin. Washing buffer with increased concentration of salt was used during transferrin purification and few randomly chosen transferrin eluates (7.5 x concentrated) per each plate were analysed by SDS-PAGE to check for potential contaminants and ensure quality of transferrin purification from isolation to isolation (Supplementary Figure 12). Moreover, transferrin eluate has been analysed for transferrin purity by performing trypsin digestion and LC-MS analysis of obtained (glyco)peptides and was shown to be 99.36% pure. We have added these details to Supplementary Materials and Methods, lines 12-26, added Supplementary Figure 12 (SDS-page of transferrin isolation from IgG depleted plasma) and added a reference to Trbojević-Akmačić, I. *et al.*¹ in lines 534-535 of the main text.

The authors indicate that not all SNPs were considered in the meta-analyses of the two cohorts. It should be specified how many SNPs were removed prior to meta-analyses as this could potentially introduce bias.

As suggested by the reviewer, we included the number of SNPs removed at each quality control step prior to the meta-analysis. This is now reflected in lines 580-583.

The authors indicate that several SNPs associating with transferrin glycosylation are classified as missense variations causing amino acid substitutions. For example, it is indicated that the FUT6 variant encodes an amino acid change that leads to an inactive enzyme. If so, it is to be expected that alpha-1,3 fucose expression on transferrin/IgG is decreased. The conclusions presented by the authors would be substantiated

considerably in case this is analysed specifically in donors expressing this variant as opposed to donors expressing two functional alleles.

We thank the reviewer for the comment. As indicated in the main text, the rs17855739 variant results in an amino-acid change that was shown by Mollicone *et al.*² to impact the enzyme activity. In their work they show that the specific amino acid substitution in the otherwise wild type *FUT6* gene results in reduced enzyme activity. While the authors test for the presence of the mutated transcript, they do not assess overall change in expression levels of the enzyme. Indeed, when checking the variant in various resources that accumulate information on genetic influence on gene expression (GTEx, eqtlGen consortia, Phenoscanner), the variant does not seem to have an impact on expression of the enzyme. We show that TfGP32 is strongly ($p\text{-value}=1.1\times 10^{-33}$) associated with rs17855739, with T allele resulting in lower levels of TfGP32. As the underlying glycan structure of the TfGP32 is currently not known, we used total plasma glycosylation GWAS to confirm that the same underlying causal variant contributes to levels of both TfGP32 and two plasma glycosylation traits containing antennary fucose (PGP32 - A4F1G3S[3,3+6,3+6]3 and PGP36 - A4F1G4S[3,3,3,6]4) from Sharapov *et al.*³ and a plasma glycosylation trait reflecting total antennary fucosylation (A-FUC) from Huffman *et al.*⁴ Transferrin is one of the most abundant proteins in plasma and it can be expected it contributes to total plasma glycosylation peaks. All tested plasma glycan peaks showed strong colocalisation within the *FUT6* region (Supplementary Figure 5), suggesting that transferrin might be contributing to these plasma glycan traits and that TfGP32 might contain antennary fucose and therefore act as a proxy for enzyme activity. These, however, warrant further investigation and experimental validation. We have now expanded on this in lines 208-220 of the main text, lines 145-154 in Supplementary Results and added Supplementary Figures 4 and 5.

If I am not mistaken, alpha1,3 fucose is hardly present in IgG1, and -as such it is surprising to note that FUT6 associates with IgG N-glycosylation. Could the authors comment on this notion and indicate where and how often this fucose is present on IgG1. As indicated above, the data conclusions would be substantiated considerably in case the authors would quantify this glycoform on IgG derived from relevant donors.

Indeed, antennary fucose is not normally found on IgG glycans. However, this association has also been observed in the discovery analysis in Klarić *et al.*⁵, consisting of the meta-analysis of 4 cohorts, out of which 1 (CROATIA-Korcula) was used in the present study, but using non-overlapping samples, and in the validation analysis of the same study (ST2 in the Klarić *et al.*⁵), using the Leiden Longevity Study where subclass specific IgG glycans were quantified using a different method, liquid chromatography coupled to a Maxis Impact quadrupole time-of-flight-MS (LC-ESI-MS/MS). As we also consistently identified a genetic association with IgG glycans at *FUT6* gene in this study, we consider this association robust. Thus, we expect *FUT6* to be indirectly associated with IgG glycosylation, through antennary fucosylation of other enzymes or proteins that participate in IgG glycosylation and have an effect on IgG GP20. We have further elaborated this in lines 409-411.

From the description it is not clear which IgG subclass is analysed. This should be specified.

We thank the Reviewer for pointing out that this information could be stated more clearly within the manuscript. N-glycans have been analysed from the total IgG pool (all subclasses). We made this clearer by rephrasing sentences at lines 531, 539-540 and 542.

The subsequent analyses assessing whether transferrin glycosylation variants associate with other complex traits add an additional complexity to the study and provide little mechanistic insight into the biology of the associations. These studies, presented in supplementary table 7, are also prone to false positive findings and should be replicated in an independent manner.

As the Reviewer rightfully pointed out, Phenoscanner (results reported in Supplementary Table 7/11a in current revision) does not have any statistical value in establishing shared genetic associations or causal relationships between traits, but was used to facilitate the cross-referencing of transferrin glycans-associated variants with the broad range of human traits. The traits that were indicated by Phenoscanner analysis to share associated variants with transferrin glycosylation, were then formally tested for sharing of the association signal with SMR-HEIDI. This method assesses whether both traits are likely to be affected by the same underlying (unobserved) causal variant or each trait is affected by a different variant in the same region. We previously reported the results of these analyses in Supplementary Table 11b. We have now also used two-sample Mendelian Randomisation (MR) to systematically evaluate the causal role of transferrin glycan traits on complex traits/diseases, and, conversely, of complex traits/diseases on transferrin glycan traits. Following the recent recommendations supporting the use of colocalisation as a follow-up analysis to reduce MR false positives⁶, we then applied an alternative approach to SMR-HEIDI - the Approximate Bayes Factor colocalisation analysis, to investigate whether the genetic regulation of both transferrin glycans and complex traits and diseases is driven by the same underlying causal variant. When possible, we performed these analyses using publicly available summary statistics of larger and more recent GWAS of complex traits (listed at Supplementary Table 18), validating these findings both by using a different statistical approach, but also different complex traits/disease data. As indicated in Supplementary table 12, MR suggests that genetically increased levels of TfGP28 are likely to affect CRP, total cholesterol and LDL, while genetically increased levels of TfGP14 increase risk for ulcerative colitis (UC). However, these results are based on few instrumental variants and were driven by a single locus (*HNF1A*). Bayesian colocalisation analysis suggests that TfGP14 and UC are regulated by distinct causal variants. The association of TfGP28 and CRP, cholesterol and LDL is driven by association in *HNF1A* locus (Supplementary Table 13), suggesting that the *HNF1A* locus is pleiotropic and has an impact on both transferrin glycan levels and complex traits.

This is now reflected in lines 246-259, 484-489 and 736-760, Supplementary Tables 12 and 13 and Supplementary Figure 6.

The authors show that IGH1 is, as expected, not expressed by hepatocytes, but expressed by plasma cells. The authors also show that these cells, but not hepatocytes, express IKZF1. Something similar is found for HNF1A and TF in hepatocytes. It is, however, unclear to me why this would suggest that variants of HNF1A/IKZF1 would be involved in regulation of FUT8 expression in liver/plasma cells without additional functional studies. Although, the authors published on IKZF1 and IgG glycosylation, such studies should be performed for hepatocytes as well to drive this case more convincingly.

We thank the reviewer for the comment. Analysis of gene knockdowns (RNAi) performed by Lauc *et al.*⁷ showed that *HNF1A* directly regulates the expression of fucosyltransferase (FUT) genes in hepatocytes (HepG2 cell line). *HNF1A* knockdown resulted in down-regulation of *FUT6* and up-regulation of *FUT8*. While it might be expected that a change in levels of *FUT6* and *FUT8* enzymes impacts levels of the product of their enzymatic activity (i.e. levels of antennary and core fucosylation), this link has yet to be experimentally proven. Since we have not investigated the effect of *HNF1A* knockdown on fucosylation of transferrin, we accordingly added more context and toned down our claims regarding the mechanism of *HNF1A* transcription factor in transferrin glycosylation, reflected in lines 440-445 and 447-450.

Reviewer #2 (Remarks to the Author - key comments indicated with **):

Lines 51-54: various associations between N-glycosylation and diseases listed. The authors should spell out that these correlations are not necessarily causal (e.g. it seems unlikely that N-glycosylation is the cause of low back pain).

As suggested by the reviewer, we made sure to clarify that the listed associations of N-glycosylation with various diseases have so far not been shown to be casual. This is now reflected in lines 56-58.

Use of ‘SNP’ throughout the manuscript – do the authors really only mean SNP ie are none of the instances referring to indels at all (which will also be included with the array genotyping + imputation strategy they described)? If they actually mean SNPs + indels then they should change to ‘SNPs’ to “genetic variants”.

We would like to thank the reviewer for giving us the opportunity to clarify this point. Only SNPs are included in the Haplotype Reference Consortium (HRC) r1.1 panel, which was used for genotype imputation of CROATIA-Korcula and VIKING cohorts. For clarity, we now included in the manuscript (at lines 102 and 573) the bibliographic reference to the HRC panel.

Line 147-8: “transferrin glycosylation variants” – I dislike the terminology as it’s potentially misleading (the reader may think the variants are all in transferrin gene region). Please say “variants associated with transferrin glycosylation” instead.

We thank the reviewer for the comment - we accordingly replaced “transferrin glycosylation variants” with “variants associated with transferrin glycosylation” throughout the text.

****Lines 151-157: the authors use the Heidi-SMR analysis to conclude the same variant underlies the glycosylation trait and the eQTL. It would be helpful to understand the causal relationship (or lack thereof) between the mRNA trait and the glycosylation trait, and the direction of any causal effect. Ie does SNP -> trait A -> trait B, or does SNP -> trait B -> trait A, or does SNP separately affect trait A and trait B, but not via each other? The paper would be strengthened by performing bidirectional MR / mediation analysis to tease this out.**

While we strongly agree with the reviewer that understanding the causal relationships between transferrin glycan traits and gene expression would be essential for unravelling the genetic mechanisms regulating glycosylation, we argue that bi-directional MR is not able in this case to robustly determine whatever (A) gene expression causes glycosylation, (B)

glycosylation causes gene expression or (C) gene expression and glycosylation are not causally related, but a genetic variant influences them each through different pathways or a confounder. One of the main assumptions of MR is that the SNP (instrumental variable - IV) associates with the outcome exclusively through the exposure (exclusion restriction assumption). When only a single region with a single cis-acting SNP is associated with both exposure and outcome, the bi-directional MR often cannot be used to determine causal direction between two traits because we cannot ensure valid instruments for both traits - SNPs associated with glycan levels used as instrumental variables should not be the same SNPs used as the instrumental variables for gene expression levels⁸. Given that the true underlying causal variant is usually not known, variants significantly associated with the trait are typically used as instruments, which in case of a colocalising genetic signal will be the same variants for both glycan levels and gene expression. Therefore, it can be expected to observe an apparently robust causal association in both directions, which does not necessarily represent the real biological mechanism behind the (possible) relationship between the two traits.

Indeed, as shown in the Revision Table 1, bi-directional MR suggests a feedback loop between glycan levels and gene-expression. To address the problem of directionality in case of single associated cis-acting variant, Hemani *et al.*⁸ proposed a Steiger test, which infers the direction from the strength of association between instrumental variables and exposure and outcome, suggesting that if the variant is more strongly associated with one of the traits, it is more likely that the causal effect is driven by that trait. We have therefore performed the Steiger test using either gene expression in different tissues or glycan levels as outcomes. As can be seen in Revision Table 1, there is no significant difference in strength of association of instrumental variable (IV) with either of the traits in SNP -> TfGP17 (exposure) -> ST3GAL4 (outcome) or SNP -> ST3GAL4 -> TfGP17 analysis (Steiger p-value > 0.05). This also indicates that neither sets of the IV satisfy the exclusion restriction assumption, as both are associated with both traits. In the case of TfGP21 and B3GAT1 variance explained by IV could be indicative of glycan levels having an impact on the gene expression in peripheral blood, however the same does not hold for expression in the liver (no difference in variance explained as estimated by Steiger p-value), where much stronger IV for gene expression is available. Given that none of these IVs satisfy the exclusion restriction assumption, and individual-level gene expression data is not available for mediation analysis, we feel that reporting these results might lead to misleading interpretation of feedback loops between protein glycosylation and gene expression. While these might be true and exist, we cannot infer this using the available data and methods.

Exposure	Outcome	MR beta	MR p-value	R2 exposure	R2 outcome	Steiger p-value
TfGP17	ST3GAL4 (GTEx liver)	-0.8942	1.45x10 ⁻¹¹	0.162	0.213	2.50x10 ⁻¹
ST3GAL4 (GTEx liver)	TfGP17	-1.0814	2.87x10 ⁻⁶²	0.225	0.137	1.32x10 ⁻¹

TfGP21	B3GAT1 (eQTLGen)	0.3386	1.12×10^{-196}	0.209	0.028	3.12×10^{-42}
B3GAT1 (eQTLGen)	TfGP21	1.8042	8.65×10^{-38}	0.033	0.115	5.74×10^{-13}
TfGP21	B3GAT1 (GTE _x liver)	-0.6967	1.82×10^{-08}	0.216	0.149	2.54×10^{-1}
B3GAT1 (GTE _x liver)	TfGP21	-1.1014	3.36×10^{-68}	0.209	0.180	6.40×10^{-1}

Revision Table 1. MR Steiger directionality test evaluating the direction of causations of transferrin glycan traits and gene expression in liver and blood. Exposure, glycan or gene expression trait tested as exposure; Outcome, glycan or gene expression trait tested as outcome; MR beta, estimate of the causal effect of the exposure on the outcome; MR p-value, p-value of the effect estimate; R² exposure, variance in the exposure explained by used instrumental variables; R² outcome, variance in the outcome explained by used instrumental variables; Steiger p-value, p-value of the direction of the causal relationship.

A similar comment applies to line 193: the authors have a similar interesting link between a variant associated with ulcerative colitis (UC) and glycosylation. Extending this to formally test whether transferrin glycosylation plays a causal role in UC via MR would greatly strengthen the potential translational value of the paper.

Given that, unlike gene expression analysis, multiple loci are significantly associated with complex traits and diseases, as suggested by the reviewer, we investigated the causal relationships between transferrin glycan traits and ulcerative colitis (and other human complex traits/diseases that colocalise with transferrin glycans according to SMR-HEIDI) by performing bi-directional two-sample Mendelian Randomisation (MR). To further corroborate our findings, we next performed Approximate Bayes Factor colocalisation analysis, an alternative approach to SMR-HEIDI analysis. The bi-directional MR suggested that genetically increased levels of glycans are more likely to influence the disease risk rather than the other way round (Supplementary Table 12). However, glycan → disease associations are based on few instrumental variants so caution must be taken when interpreting these findings. Bayesian colocalisation analysis has showed that in the case of TfGP14 and UC there are two distinct variants in the NXEP1/4 region affecting glycosylation and disease risk, while the effect of TfGP28 on LDL, cholesterol and CRP is driven by the association in *HNF1A* locus (Supplementary table 13). This locus encodes transcription factor HNF1A, suggesting that this association is pleiotropic and has an impact on both transferrin glycan levels and complex traits. This is now reflected in lines 246-259, 484-489 and 736-760, Supplementary Tables 12 and 13 and Supplementary Figure 6.

****How many of the individuals in the cohort had diabetes? Have the authors considered whether tendency to diabetes or “pre diabetes”/ relative insulin resistance (or even just different glucose levels with the continuum of normal) might have impacted their findings?**

This could be addressed in a couple of ways:

- I. Do the loci associated with glycosylation overlap with diabetes susceptibility loci (T2 probably most relevant given its prevalence) or loci associated with HbA1c levels?**
- II. If the authors have measures of glycaemia available (eg HbA1c contemporaneous with plasma sample), they could add this as a covariate to see whether this attenuates any of the loci? If so, they could do a formal mediation analysis to test the hypothesis that the effect is mediated via blood sugar levels at the relevant locus.**

We investigated whether prediabetes status, insulin resistance or unusual blood glucose levels might have impacted the identified genetic associations with transferrin glycans, in the two ways suggested by the reviewer.

(I) By searching on Phenoscanner, a curated database holding publicly available results from large-scale GWAS, we did not find any overlap between loci associated with transferrin glycans and loci associated with diabetes, insulin resistance or HbA1c levels (Supplementary Table 11a).

(II) We re-ran our transferrin glycosylation GWAS for VIKING cohort, adding HbA1c levels as a covariate. Unfortunately, HbA1c levels were not available for CROATIA-Korcula cohort, but we expect that the outcome of the analysis would be similar given that both cohorts represent a general population with similar characteristics. By adjusting for HbA1c levels, we obtained the same significant loci-glycan trait associations as in our original GWAS (where glycan traits were also adjusted for age, sex and relatedness). As reported in Revision Table 2, the association effect sizes and p-values obtained in the GWAS adjusted for HbA1c levels are very similar to those obtained in the original GWAS. Given the results of our test, we could not find any evidence suggesting that prediabetes, insulin resistance or HbA1c levels might have impacted our findings. We have therefore removed the speculation over diabetes from the manuscript.

Locus	Gene	SNP	EA	EAf	Lead glycan	Beta	Beta HbA1c	Se	Se HbA1c	P	P HbA1c
2:134854659-135016618	MGAT5	rs2442046	C	0.79	TfGP23	-0.457	-0.459	0.055	0.055	3.19x10 ⁻¹⁶	1.63x10 ⁻¹⁶
11:126019570-126264155	ST3GAL1	rs4307732	A	0.107	TfGP17	0.825	0.835	0.067	0.066	2.25x10 ⁻³²	9.83x10 ⁻³⁴

11:134256805 -134294813	B3GAT4	rs78760579	G	0.11	TfGP21	-0.803	-0.810	0.069	0.069	1.84x10 ⁻²⁹	2.78x10 ⁻³⁰
14:65789571 -66221771	FUT8	rs1113962	T	0.75	TfGP20	0.476	0.475	0.051	0.051	4.43x10 ⁻²⁰	5.1x10 ⁻²⁰
19:5813766 - 5846277	FUT6	rs79008529	A	0.037	TfGP32	-1.029	-1.028	0.121	0.121	6.16x10 ⁻¹⁷	6.29x10 ⁻¹⁷

Revision Table 2. Comparison of transferrin glycans association signals in VIKING cohort with and without adjustment for HbA1c levels. Locus - coded as “chromosome: locus start–locus end” (GRCh37 human genome build); Gene - suggested candidate gene; SNP - variant with the strongest association in the locus; EA - SNP allele for which the effect estimate is reported; EAF - frequency of the effect allele; Lead glycan - glycan trait with the strongest association to the reported SNP; Beta - effect estimate for the SNP and glycan with the strongest association in the locus; Beta HbA1c, effect estimate for the SNP associated with the glycan conditioned by HbA1c levels; SE, standard error of the effect estimate; SE HbA1c, standard error of the effect estimate conditioned by HbA1c levels; P, p-value for the effect estimate; P HbA1c, p-value for the effect estimate conditioned by HbA1c levels.

Lines 159-171. The authors identify some instances where a protein-coding variant is in high LD with the variant associated with glycosylation. What was the situation for the ‘cis’ acting variant at *TF* (encoding transferrin itself)? If there is no protein coding variant in high LD with the lead variant, then can the authors comment on how it might be impacting glycosylation (as transferrin protein structure would presumably not be affected?). Is this variant a cis eQTL or pQTL for TF?

We thank the reviewer for raising this interesting point. The variant most strongly associated with transferrin glycosylation at *TF* gene, rs6785596, is not in LD with any transferrin pQTLs.

To assess the potential impact of transferrin protein levels on transferrin glycome associations we used transferrin cis-protein QTL (pQTL) rs8177240 (LD $R^2 = 0.02$ with the glycan QTL, glyQTL rs6785596), the strongest association with transferrin protein levels reported in GWAS catalog (p-value = 8×10^{-610})⁹, as a proxy for TF abundance. Interestingly, the pQTL is not an expression QTL (eQTL), but rather a splicing QTL for transferrin levels in liver (p= 5.9×10^{-25} , GTEx v8). We then tested its association with transferrin glycans and assessed whether the glycan associations with variants from the *TF* region are likely to be driven by this variant. We considered four models:

- M0: glycan ~ age + sex
- M1: glycan ~ age + sex + pQTL (rs8177240)
- M2: glycan ~ age + sex + glyQTL (rs6785596)

M3: glycan ~ age + sex + pQTL (rs6785596) + glyQTL (rs8177240)

and performed likelihood ratio test between:

- M0 and M1 to assess associations of glycans and pQTL (rs8177240)
- M1 and M3 to assess whether glyQTL contributes to glycan levels even when the pQTL is included in the model
- M2 and M3 to assess whether pQTL contributes to glycan levels even when the glyQTL is included in the model

Two glycan traits were significantly ($p \leq 0.05/35 = 1.4 \times 10^{-3}$) associated with the pQTL. For one of the two traits, TfGP3, the glyQTL contributes to glycan levels even when the pQTL is included in the model, while for the TfGP9 no additional variation is explained by the glyQTL (Supplementary Table 6).

To further corroborate these findings we also repeated the meta-analysis conditioning on the transferrin pQTL rs8177240⁹. The only glycan trait that showed a relevant change in effect size and significance of its association was TfGP9 (Revision Table 3), suggesting that its association was dependent on the transferrin protein levels. In case of two glycan traits, TfGP3 and TfGP8, the associations were somewhat less significant, but the effect sizes remained very similar. Accordingly, we consider that transferrin protein levels are likely not affecting associations with 2 out of 3 transferrin glycan traits. This is in accordance with findings from Kutalik *et al.*¹⁰, who used the same approach to show that associations of α -disialylated transferrin with the *TF* region were independent of associations with transferrin pQTL.

The sentinel glycosylation variant, rs6785596, is a cis eQTL in adipose tissue (Supplementary Table 8a) and it colocalises with TfGP3, but no colocalisation is observed between TfGP3 and *TF* expression in blood (Supplementary Figure 3). However, as outlined in the main text, transferrin is predominantly expressed in liver, for which there are no robust transferrin eQTLs (the strongest eQTL in GTEx v8 rs60770862, $p = 3.3 \times 10^{-6}$, LD with glyQTL rs6785596 = 0.0001). The glyQTL variant rs6785596 is also in mild LD (0.57) with a missense variant rs1799899. Altogether, further analyses are needed to unravel the complex mechanism behind these associations. Interestingly, similar association were also observed for IgG glycosylation, with some IgG glycan traits associating with variants in the *IGH* locus, coding for heavy chain of IgG. Exact mechanism how these “cis-gly”-QTLs could be affecting glycosylation levels remains unclear. This is now reflected in lines 127-137, 480-484 and 608-631 of the main text, 156-229 of the Supplementary Results, Supplementary Table 6 and Supplementary Figure 3.

SNP	SNP_id	EA	EAF	Glycan	Beta	Beta C	SE	SE C	P	P C
-----	--------	----	-----	--------	------	--------	----	------	---	-----

rs6785596	3:133466457	A	0.047	TfGP3	0.7870	0.7442	0.075	0.078	1.57x10 ⁻²⁵	1.02x10 ⁻²¹
rs6785596	3:133466457	A	0.047	TfGP8	0.7263	0.6886	0.075	0.077	1.78x10 ⁻²²	2.23x10 ⁻¹⁹
rs9830001	3:133433470	A	0.384	TfGP9	-0.2125	-0.0864	0.032	0.033	4.35x10 ⁻¹¹	8.00x10 ⁻³

Revision Table 3. Transferrin TfGP3 and TfGP8 glycans association signals at chromosome 3 before and after conditioning by transferrin pQTL. SNP, variant with the strongest association in the locus for the glycan trait; SNP_id, SNP location coded as "chromosome:base pair position" according to GRCh37 human genome build; EA - SNP allele for which the effect estimate is reported; EAF - frequency of the effect allele; Glycan, glycan trait associated with the reported SNP; Beta, effect estimate for the SNP associated with the glycan; Beta C, effect estimate for the SNP associated with the glycan conditioned by transferrin pQTL; SE, standard error of the effect estimate; SE, standard error of the effect estimate conditioned by transferrin pQTL; P, p-value for the effect estimate; P, p-value for the effect estimate conditioned by transferrin pQTL.

****Is glycosylation analysed as a quantitative measure or a ratio of glycosylated to unglycosylated? If the former, then simply increasing total protein abundance might lead to a higher amount of glycosylated protein without changing the ratio. The authors should clarify whether their approach has taken this into consideration as this will have an important bearing on the interpretation of results (analogous to looking at absolute vs proportional cell counts in a full blood count).**

Glycosylation was analysed by releasing total N-glycans from the isolated protein and each glycan structure was quantified as the percentage of the total IgG N-glycome or total transferrin N-glycome. As it is detailed in lines 547-551, the raw glycan measurements are total area normalised. By using this methodological approach, only changes in glycosylation are detected (relative abundance of individual glycan species in relation to the whole IgG or transferrin glycome) and not the absolute amounts of specific glycans, which would be affected by changes in protein expression. Unfortunately, protein abundance was not measured for either transferrin and IgG in this study and, consequently, the relation between protein abundance and N-glycan measurements could not be directly investigated. In the case of IgG however, we checked whether some specific protocols used for glycan analysis have a bias depending on the initial, already isolated, protein amount. The protocol we used was robust in measuring very similar levels of glycan traits for different IgG abundance (Supplementary Figure 13). In addition, Bermingham *et al.*¹¹ reported that, while there were some associations between IgG N-glycan traits and IgG levels, adjusting for IgG levels in the analyses made no meaningful difference to associations of glycans with markers of glycaemic control. To assess the potential impact of transferrin protein levels on transferrin glycome, we used transferrin pQTL as a proxy for protein levels and showed that two glycans might be

affected by protein levels, but for one of the glycans the glyQTL (the sentinel variant in the *TF* region) contributes to levels of the glycan in addition to the pQTL.

Supplementary Figure 13: Levels of IgG glycan traits (GP1-24) measured for different initial amounts of isolated IgG protein.

In accordance with what was observed for IgG glycome, transferrin protein levels might have an overall minor effect on transferrin glycome levels, most likely on two out of three glycans associated with the *TF* region, but not such to result in major differences in reported associations. We provide a more detailed answer in the reply to the previous question “Lines 159-171. The authors identify some instances where a protein-coding variant is in high LD with the variant associated with glycosylation.”. We have included this topic in the lines 127-137, 480-484 and 608-631 of the main text, 156-229 of the Supplementary Results, Supplementary Table 6 and Supplementary Figures 3 and 13.

Line 176 – ‘TF’ abbreviation for transcription factor. Please avoid this since it creates confusion with the *TF* gene discussed elsewhere!

We agree with the reviewer that having the same abbreviation (TF) referring to two different instances (transferrin gene and transcription factor) might be confusing for the reader. We accordingly removed all instances of “TF” abbreviation used for transcription factor. This is reflected in lines 224-225 and 669-670 of the main text and in the header of Supplementary Table 10.

Figure 2A and 2B, and Figure 4A and B would be better presented vertically so the reader can more easily see how the signals compare.

Thank you for the suggestion, we edited Figure 2 and Figure 4 by rearranging A and B panels vertically.

The authors rightly discuss the importance of PTMs in their many various contexts. The Discussion should highlight that this study is limited to plasma proteins i.e. extracellular space. Some of the PTMs discussed in the manuscript (e.g. phosphorylation) play an important role in intracellular signalling with dynamic on/off effects. The impact of genetics on these types of PTMs will be very challenging to assess, but will be an interesting area for future research.

We would like to thank the reviewer for this interesting observation. We have included this point in the manuscript discussion, at lines 502-506.

Lines 388-397: I found this a bit speculative even for Discussion and felt it would be better removed. Ultimately as the authors point out “However, this SNP has so far not been associated with diabetes or diabetes-related traits, suggesting that this relationship needs to be explored further”. Without this confirmatory link, the point is not particularly convincing.

Thank you for suggestions, we have removed this from the discussion.

Lines 402-403 “glycosylation SNPs in NXPE1/NXPE4 locus were pleiotropic with ulcerative colitis” – I found the phrasing odd. I realise the terminology comes from SMR HEIDI but it is misleading language. A variant associated with a single disease and a molecular trait can hardly be considered pleiotropic in the usual sense. Disease-associated variants must have molecular underpinnings and so association with some molecular trait is inevitable (even if yet to be discovered). I would reserve the term ‘pleiotropic’ for when the variant is associated with multiple traits within the same trait type (eg with multiple proteins, or with multiple diseases). The authors should rephrase this more simply to say what they mean ie “variants in NXPE1/NXPE4 locus associated with glycosylation were also associated with ulcerative colitis”. There were a couple of other instances where pleiotropy was referred to that would similarly benefit from clearer language. The other reason for doing this is to avoid confusing readers more familiar with MR, where horizontal pleiotropy violates MR assumptions and thus calls into question the conclusions of an MR analysis.

We thank the reviewer for highlighting this potential source of confusion. We replaced the term “pleiotropy” with “colocalisation” throughout the text (lines 161, 194, 239, 242 and 668).

Line 502 Locus Definition. I found the explanation confusing. Clarifying or providing pseudocode or real code might help understand what was done in terms of grouping loci across different glycosylation traits.

To make the locus definition procedure clearer, we included the Supplementary Figure 11, visually representing the different steps taken to perform the described procedure and supporting the textual explanation.

****To maximise utility of the results to the community, the authors should ensure full summary stats are uploaded to an appropriate repository (eg GWAS catalog).**

We appreciate the need to share our results with the scientific community, especially since this is the first GWAS of transferrin glycome. Summary statistics are available at the Edinburgh DataShare - a digital repository of research data produced at the University of Edinburgh (<https://datashare.ed.ac.uk/handle/10283/4059>). We will also upload our results to GWAS catalog after publication. This is now reflected in lines 1019-1022.

Reviewer #3 (Remarks to the Author):

It is at times difficult to follow the different glycan traits, because the data is virtually always collapsed into strongest association. This perhaps makes sense in the main text and tables, but particularly in supplementary tables, I recommend providing a complete overview of all glycan traits with all independent variants (i.e. all GCTA-COJO independent associations to all measured glycan traits), which could help with subsequent two points as well. Alternatively please provide some explanation or quantification of differences in glycan traits.

As suggested by the reviewer, we edited Supplementary Table 5 to include all significant (p-value $< 1.43 \times 10^{-9}$ for transferrin and $< 2.08 \times 10^{-9}$ for IgG) SNPs independently associated with each transferrin and IgG glycan trait.

Please provide more details regarding the findings in each of the two individual cohorts, e.g. calculate a heterogeneity metrics or list effect size and P-value per cohort, and summarize this in main text. Supplementary figure 5 and 6 are very useful towards this, and supplementary table 13 also serves some of this purpose, but it is for example not possible to know all 10 findings of e.g. table 1. Also, it took me a while to figure out the high level of replication between the two cohorts, which is why I think it could be highlighted more.

As suggested by the reviewer, we included in the main text more details about single cohort transferrin glycans GWAS results and their replication, details of which are now in lines 103-115. Since the topic is now reported in the main text, we accordingly removed the corresponding text in the Supplementary Results.

Please provide calculations of proportion of variance explained for each association and discuss the overall proportion of variance explained by genetics for each trait. This is currently only provided for top-SNP per locus (table 1), not all independent SNPs. It will be of interest to readers to know to what degree these traits are genetically regulated overall. Also consider adding in LDSC calculations, or at least discussing possibility of low-effect background.

As suggested by the reviewer, we included in Supplementary Table 5 (now edited to include all significant SNPs independently associated with each transferrin and IgG glycan trait) the percentage of phenotypic variance explained by all independently associated SNPs. We agree with the reviewer that it will be of interest to know to what degree transferrin glycan traits are genetically regulated. Due to the small sample size, we were unable to apply the most commonly used methods for calculating heritability using both GWAS summary statistics and individual genotype data [i.e. linkage disequilibrium score regression (LDSC), High-definition likelihood inference of genetic correlations (HDL) and GCTA genomic-relatedness-based restricted maximum-likelihood (GREML)] and obtaining reliable heritability estimates. We thus estimated cohort-specific heritability for each transferrin glycan trait by using the “polygenic” function from the “GenABEL” R package¹², as reflected at lines 109-113 and 662-663. We report the results of this analysis in Supplementary Table 3.

The study is based on isolated populations, which may be in some contexts be considered a strength. However, the GCTA-COJO needs a standardized LD value source, which is indicated as being from the UK biobank. Will this affect the results, in case the LD patterns are different? Was this tested?

As the reviewer correctly observed, identifying the most suitable LD reference sample is a key choice for performing GCTA-COJO analysis. GCTA-COJO’s authors advise against using a reference panel with a sample size < 4000 (see Supplementary Figure 4 of Yang, J. *et al.*¹³), such as HapMap or 1000G. We thus used UKBB, since it represented a suitable choice, in terms of sample size and ancestry for our European-heritage cohorts (one Scottish, one Croatian). Moreover, the conditional analysis was performed on meta-analysis summary statistics, which by their nature average out the impact of isolated populations. Since we do not have access to another reference panel of the suitable size, we could not test its potential effect on the conditional analysis results. However, we acknowledge this limitation in lines 650-652.

At lines 105-109, there are 6 genes with independent contributions listed: ST4GAL4, MGAT5, B3GAT, FUT8, FUT6 and TF. The numbers listed are 7, 4, 4, 2, 2 and 2. In addition to the remaining 4 genes, adding up to 10 loci with significant variants. But that sums up to only 25 - not 26 independently contributing variants, stated in line 100. Please double-check that reported counts are correct.

We thank the reviewer for noticing this inconsistency that highlighted a mistake we did in counting the independently contributing variants. We were counting different variants from the same region associated by multiple glycan traits as independent traits, while some of them were in high LD. We therefore changed this to count only the maximum number of variants from the region associated with a single trait, which reduced the number of variants to 15. This correction is reflected in lines 118 and 123-127 of the main text.

The study by Kotalik et al 2011, PMID 21665994, “Genome-wide association study identifies two loci strongly affecting transferrin glycosylation” is related to this manuscript. The study is already briefly mentioned in supplementary table 7a, but the manuscript could benefit from more discussion and comparison with such a related study.

We thank the reviewer for raising this important point. To quantify transferrin glycosylation Kotalik *et al.*¹⁰ have used capillary electrophoresis and immunoassays to specifically measure a composite glycan trait that consists of transferrin glycans without sialic acids (asialotransferrin) and glycans with 2 sialic acids (disialotransferrin), referred to as carbohydrate-deficient transferrin (CDT). This relatively crude measure gives a partial insight into sialylation status of the protein and has been used as a biomarker of alcohol consumption. The authors found 2 regions associated with the trait, *PGMI* and *TF*, one of which we also find in the current work (*TF*). In our study we were able to measure 35 different transferrin glycan traits, providing higher resolution of underlying structures, which resulted in greater power to detect biologically meaningful associations. We also note that since the *TF* region was already shown to be associated with transferrin glycosylation, we cannot consider it as a novel finding so we reduced the number of new loci from 10 to 9. We have now included this in the abstract and in lines 386-392 of the main text.

I think it would be more useful to readers with the very big supplementary tables in a more data-readable format than pdf (e.g. xlsx or txt).

We completely agree with the reviewer that providing a spreadsheet of our larger Supplementary Tables would be more useful for the readers. However, the xlsx format was not accepted in the manuscript submission process and we had to convert our files to pdf. However, we will ask the editor whether it would be possible to provide the xlsx or at least the txt version of the Supplementary Tables.

In Supplementary Table 7a the first two columns are completely identical, except the header name; snp and ref_rsids. What's the motivation to provide that twice? If none, consider removing

We agree with the reviewer that these two columns, carrying the same type of information, represent an unnecessary redundancy. We accordingly removed the "ref_rsids" column from Supplementary Tables 8a and 11a (former 7a), both reporting Phenoscanner results.

Revision references

1. Trbojević-Akmačić, I. *et al.* Chromatographic monoliths for high-throughput immunoaffinity isolation of transferrin from human plasma. *Croat. Chem. Acta* **89**, 203–211 (2016).
2. Mollicone, R. *et al.* Molecular basis for plasma $\alpha(1,3)$ -fucosyltransferase gene deficiency (FUT6). *J. Biol. Chem.* **269**, 12662–12671 (1994).
3. Sharapov, S. Z. *et al.* Defining the genetic control of human blood plasma N-glycome using genome-wide association study. *Hum. Mol. Genet.* **28**, 2062–2077 (2019).
4. Huffman, J. E. *et al.* Polymorphisms in B3GAT1, SLC9A9 and MGAT5 are associated with variation within the human plasma N-glycome of 3533 European adults. *Hum. Mol. Genet.* **20**, 5000–5011 (2011).
5. Klarić, L. *et al.* Glycosylation of immunoglobulin G is regulated by a large network of genes pleiotropic with inflammatory diseases. *Sci. Adv.* **6**, eaax0301 (2020).
6. Zheng, J. *et al.* Phenome-wide Mendelian randomization mapping the influence of the plasma proteome on complex diseases. *Nat. Genet.* **52**, 1122–1131 (2020).
7. Lauc, G. *et al.* Genomics Meets Glycomics—The First GWAS Study of Human N-Glycome Identifies HNF1 α as a Master Regulator of Plasma Protein Fucosylation. *PLoS Genet.* **6**, e1001256 (2010).
8. Hemani, G., Tilling, K. & Davey Smith, G. Orienting the causal relationship between imprecisely measured traits using GWAS summary data. *PLoS Genet.* **13**, e1007081 (2017).
9. Benyamin, B. *et al.* Novel loci affecting iron homeostasis and their effects in individuals at risk for hemochromatosis. *Nat. Commun.* **5**, 4926 (2014).
10. Kutalik, Z. *et al.* Genome-wide association study identifies two loci strongly affecting transferrin glycosylation. *Hum. Mol. Genet.* **20**, 3710–7 (2011).
11. Bermingham, M. L. *et al.* N-glycan profile and kidney disease in type 1 diabetes. *Diabetes Care* **41**, 79–87 (2018).
12. Karssen, L. C., van Duijn, C. M. & Aulchenko, Y. S. The GenABEL Project for statistical genomics. *F1000Research* **5**, (2016).
13. Yang, J. *et al.* Conditional and joint multiple-SNP analysis of GWAS summary statistics identifies additional variants influencing complex traits. *Nat. Genet.* **44**, 369–375 (2012).

Reviewers' Comments:

Reviewer #1:

Remarks to the Author:

Overall, the answers provided by the authors to my questions on the previous version of the manuscript are, in my opinion, not convincing.

Question 1 of previous report: To show the purity of the transferrin used, I asked for a SDS-page silverstain to assess the purity of the transferrin fraction analysed. Instead, the authors show a gel in which the protein bands were visualized with a less sensitive staining method. The authors also performed a mass-spec analyses to show purity, but mass spec is not a quantitative methods as such. Thus, even though a silver stain is quite easy to perform, it has not been done and therefore this question is not addressed convincingly.

Question 3 of previous report. It is mentioned by the authors that the FUT6 variant encoded an amino acid change that leads to an inactive enzyme. For this reason, I indicated that "If so, it is to be expected that alpha-1,3 fucose expression on transferrin/IgG is decreased. The conclusions presented by the authors would be substantiated considerably in case this is analysed specifically in donors expressing this variant as opposed to donors expressing two functional alleles"

In the rebuttal the authors indicate that the variant does not associate with FUT6 gene expression. However, this was not the question as the question related to the expression of alpha-1,3 fucose on transferrin. Although the authors speculate that TfGP32 might contain antennary fucose and therefore might act as a proxy for enzyme activity, it is also mentioned that "as the underlying glycan structure of the TfGP32 is currently unknown,....." Therefore, the question has, in opinion, not been addressed in a convincing manner leaving the FUT6/glycan trait association with a lack of functional evidence.

Question 4 of previous report; "alpha1,3 fucose is hardly present in IgG1, and -as such it is surprising to note that FUT6 associates with IgG N-glycosylation. Could the authors comment on this notion and indicate where and how often this fucose is present on IgG1. As indicated above, the conclusions would be substantiated considerably in case the authors would quantify this glycoform on IgG derived from relevant donors"

Such quantification has not been performed and the authors now speculate FUT6 to be indirectly associated with IgG glycosylation. No mechanistic insights on how this association might be explained are provided.

Question 6 of previous report: "These studies, presented in supplementary table 7, are also prone to false positive findings and should be replicated in an independent manner."

Although additional analyses on the same dataset has been performed, no independent replication has been performed. Therefore, I do not consider this question to be addressed in a convincing manner.

Question 7 of previous report "...It is, however, unclear to me why this would suggest that variants of HNF1A/IKZF1 would be involved in regulation of FUT8 expression in liver/plasma cells without additional functional studies. Although, the authors published on IKZF1 and IgG glycosylation, such

studies should be performed for hepatocytes as well to drive this case more convincingly”

The authors did modify the text of the manuscript on this aspect to some extent, but did not perform these studies.

Together, the manuscript describes that glycosylation of IgG and transferrin is under genetic regulation, but no mechanistic insights are provided and several claims made by the authors are, in my opinion, not convincingly supported by the data provided. The observation that genetic regulation influences N-glycosylation of IgG has been presented before.

Reviewer #2:

Remarks to the Author:

The authors have very thoroughly addressed my review points through new analyses and a clear and comprehensive rebuttal letter. The manuscript is now in excellent shape and is suitable for publication.

One minor issue regarding my previous comment:

"Lines 159-171. The authors identify some instances where a protein-coding variant is in high LD with the variant associated with glycosylation. What was the situation for the „cis“ acting variant at TF (encoding transferrin itself)? If there is no protein coding variant in high LD with the lead variant, then can the authors comment on how it might be impacting glycosylation (as transferrin protein structure would presumably not be affected?). Is this variant a cis eQTL or pQTL for TF?"

The authors have very comprehensively addressed the question of eQTL/pQTL and have gone over and above the call of duty with some very nice analyses modelling the effect of including the pQTL (which was not in LD with the glyco QTL) in the model.

I think the first point in my question has been overlooked, which was simply whether there was a protein-coding variant in strong LD with the lead glyco QTL?

I have taken the liberty of having a look: rs6785596, the transferrin glyco QTL in the TF region is not in strong ($r^2 > 0.8$) with any protein-coding SNPs. However, there is a missense variant, rs1799899, in moderate LD (r^2 0.57 in Europeans) with rs6785596.

It might be worth testing whether conditioning on this variant abrogates the glyco QTL, or at least mentioning that there is a missense variant that *might* provide a potential mechanistic explanation underpinning the glyco pQTL.

I do not need to review the manuscript again before publication.

Reviewer #3:

Remarks to the Author:

The revised supplementary tables, particularly 5, addresses several of my comments very well and overall, the authors have made a good job of addressing my concerns. However, there's still several minor errors in the updated tables that need to be corrected before publication:

* supplementary table 1 caption lists a PVE column, but there is none. It is probably the column

named Phe.var. (Possibly supp table 2 as well)

* supplementary table 5 caption lacks description of several columns, e.g. Gene and Phe.var

* supplementary table 4, there's reporting of effect sizes but no indication of effect allele and other allele

* There are several other minor nuisances in the tables that could be improved at the authors own discretion, i.e. why have sub-suffix numbering (S8a / S8b), why sometimes report only EA and other times both EA and OA, consistent spelling of repeated column-headers (e.g. p / P).

All other comments have been fully addressed and the authors should be applauded for this big work. I can recommend this manuscript for publication.

Reviewer #2 (Remarks to the Author):

One minor issue regarding my previous comment: "Lines 159-171. The authors identify some instances where a protein-coding variant is in high LD with the variant associated with glycosylation. What was the situation for the „cis“ acting variant at TF (encoding transferrin itself)? If there is no protein coding variant in high LD with the lead variant, then can the authors comment on how it might be impacting glycosylation (as transferrin protein structure would presumably not be affected?). Is this variant a cis eQTL or pQTL for TF?"

I think the first point in my question has been overlooked, which was simply whether there was a protein-coding variant in strong LD with the lead glc QTL? I have taken the liberty of having a look: rs6785596, the transferrin glyc QTL in the TF region is not in strong ($r^2 > 0.8$) with any protein-coding SNPs. However, there is a missense variant, rs1799899, in moderate LD (r^2 0.57 in Europeans) with rs6785596. It might be worth testing whether conditioning on this variant abrogates the glyc QTL, or at least mentioning that there is a missense variant that *might* provide a potential mechanistic explanation underpinning the glyc pQTL.

We would like to thank the reviewer for this suggestion. We have already included this in the discussion (lines 421-422) but have further highlighted it as potential alternative mechanism for the observed association.

Reviewer #3 (Remarks to the Author):

There's still several minor errors in the updated tables that needs to be corrected before publication:

*** supplementary table 1 caption lists a PVE column, but there is none. It is probably the column named Phe.var. (Possibly supp table 2 as well)**

We now fixed the inconsistencies in column names between the legend and the table in Supplementary Table 1 and 2.

*** supplementary table 5 caption lacks description of several columns, e.g. Gene and Phe.var**

We added the columns missing descriptions in the legend of Supplementary Table 5.

*** supplementary table 4, there's reporting of effect sizes but no indication of effect allele and other allele**

We included a column reporting the effect allele in Supplementary table 4.

*** There are several other minor nuisances in the tables that could be improved at the authors own discretion, i.e. why have sub-suffix numbering (S8a / S8b), why sometimes report only EA and other times both EA and OA, consistent spelling of repeated column-headers (e.g. p / P).**

We made sure to spell column-headers repeated across multiple tables consistently, as suggested.